# Ethylene industrial emitters seen from space

Bruno Franco [1] ✉, Lieven Clarisse[1], Martin Van Damme[1,2], Juliette Hadji-Lazaro[3], Cathy Clerbaux [1,3] & Pierre-François Coheur[1]

Volatile organic compounds are emitted abundantly from a variety of natural and anthropogenic sources. However, in excess, they can severely degrade air quality. Their fluxes are currently poorly represented in inventories due to a lack of constraints from global measurements. Here, we track from space over 300 worldwide hotspots of ethylene, the most abundant industrially produced organic compound. We identify specific emitters associated with petrochemical clusters, steel plants, coal-related industries, and megacities. Satellite-derived fluxes reveal that the ethylene emissions of the industrial sources are underestimated or missing in the state-of-the-art Emission Database for Global Atmospheric Research (EDGAR) inventory. This work exposes global emission point-sources of a short-lived carbonated gas, complementing the ongoing large-scale efforts on the monitoring of inorganic pollutants.

The past decade saw major breakthroughs in the detection and monitoring from space of point-sources of atmospheric pollutants, such as nitrogen dioxide ($NO_2$), sulfur dioxide ($SO_2$), and ammonia ($NH_3$)[1–3], and of greenhouse gases, e.g., methane ($CH_4$)[4]. These were achieved by taking full advantage of the high spatial and temporal sampling offered by nadir-viewing, polar-orbiting satellite instruments, combined with oversampling techniques. Such advances in remote sensing are crucial for improving gas emission inventories and for mitigating polluting releases to the atmosphere. In this respect, a clear identification of the individual gas emitters is a prerequisite. Ethylene (ethene, $C_2H_4$) is a volatile organic compound (VOC) rapidly degraded in the atmosphere close to its sources, where it contributes to air pollution as a high-yield precursor of formaldehyde and tropospheric ozone[5–9]. Although locally it can be emitted in vast quantities by biomass burning[10,11], its background concentration is dominated by natural sources and remains mostly below 0.1 part per billion (ppb) in the global troposphere[12–14]. Hence, ethylene has only been measured from space in concentrated fire plumes so far[15–18]. However, it emanates also from heavy industries and other human activities, for instance from incomplete combustion of fossil fuels and biofuels. Moreover, important releases occur from its industrial production. Ethylene is indeed the mainstay of the modern chemical industry[12,19–21] with a continuously growing production capacity of 150–180 Mt yr$^{-1}$ as it serves as the principal building block for myriad products including plastics and polymers[20,21]. For these reasons, ethylene is a unique tracer

of anthropogenic VOC emissions and air pollution related to reactive carbonated pollutants.

In this work, we expose over 300 anthropogenic $C_2H_4$ hotspots from space, captured by the Infrared Atmospheric Sounding Interferometer (IASI) satellite measurements[22], and identify different categories of hotspots related to heavy industries and megacities. We compare the satellite-derived emission fluxes of industrial $C_2H_4$ point-sources with EDGAR, showing that these fluxes are unpredicted or misrepresented in the inventory.

## Results
### Ethylene hotspots
The IASI dataset used here consists of thirteen years (2008–2020) of daily global, cloud-free, hyperspectral infrared observations[23]. For each spectrum, we calculated a hyperspectral range index (HRI) that quantifies the $C_2H_4$ signal strength with a high sensitivity. Subsequently, we converted the HRI to gas column abundance by means of an artificial neural network ("Methods"). To take advantage of the extensive IASI time series, we applied a wind-adjusted super-resolution technique to the HRI dataset[24], which allows increasing the spatial resolution of satellite data beyond the native resolution of the sounder ("Methods"; Supplementary Figs. 1, 2). By doing so, we produced a 13-year averaged global distribution of ethylene at a 0.01° × 0.01° spatial resolution (Fig. 1). The HRI is used here as it directly expresses the integrated information on the target gas contained in the satellite

[1]Université libre de Bruxelles (ULB), Spectroscopy, Quantum Chemistry and Atmospheric Remote Sensing (SQUARES), Brussels B-1050, Belgium. [2]Royal Belgian Institute for Space Aeronomy (BIRA-IASB), Brussels, Belgium. [3]LATMOS/ IPSL, Sorbonne Université, UVSQ, CNRS, Paris, France. ✉e-mail: bruno.franco@ulb.be

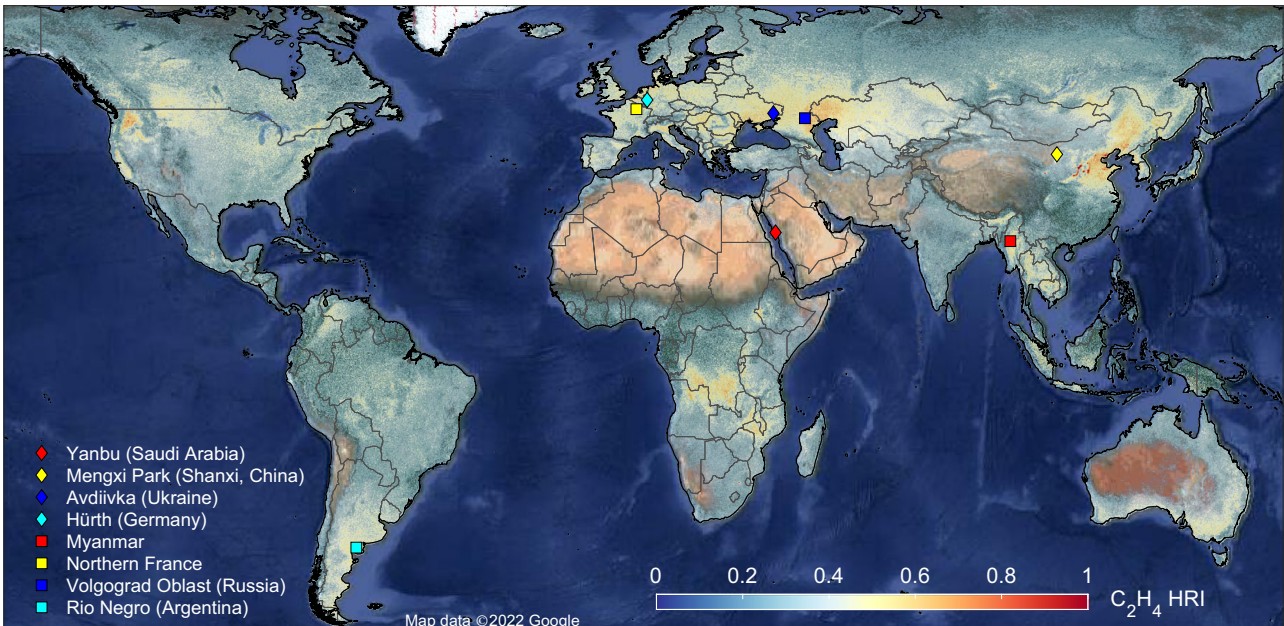

**Fig. 1 | IASI hyperfine resolution distribution of ethylene.** 13-year average (2008–2020) of $C_2H_4$ HRI from IASI obtained by wind-rotated supersampling. HRI hyperspectral range index; IASI Infrared Atmospheric Sounding Interferometer. The markers give the location of $C_2H_4$ hotspots or regions of interest that underwent a thorough spectral analysis presented in Supplementary Fig. 5. The satellite visible imagery in background is from Google Earth, CNES/Airbus, DigitalGlobe, and Landsat/Copernicus. Map data ©2022 Google.

measurement, and as it is a less noisy indicator of local gas enhancement than the retrieved column (Supplementary Fig. 3). The $C_2H_4$ global distribution shows a variable background, with regionally higher values that we attribute, via thorough IASI spectra analysis, to ethylene over polluted environments (e.g., Europe), and to surface emissivity effects over certain types of soils ("Methods"). Zoom-ins over various regions of the global map (Fig. 1) unveil—thanks to its hyperfine spatial resolution—large local $C_2H_4$ enhancements with typically a spatial extent of 20–50 km. Examples are shown over Iran and the Persian Gulf in Fig. 2, and over an industrial valley of the Shanxi Province, China, in Supplementary Fig. 4. We provide firm spectroscopic evidence that ethylene is the dominant contributor to such enhancements detected throughout the globe, confirming that those correspond to $C_2H_4$ hotspots ("Methods"; Supplementary Fig. 5). Figure 2 and Supplementary Fig. 4 depict close-up views on some of these hotspots with visible satellite imagery, demonstrating that these point-sources can be traced back primarily to heavy industries and urban areas.

**Point-sources identification**

Through careful analysis of the hyperfine resolution map, we discovered a total of 336 $C_2H_4$ hotspots worldwide. With the help of satellite visible imagery similar to what we illustrate with Fig. 2, and information collected online (e.g., on the companies and type of activities), we pinpoint for each hotspot the likely emitters of ethylene and show that in most cases they belong to three specific types of industry: (1) chemistry and petrochemistry, (2) coal exploitation and processing, and (3) metallurgy. Interestingly, some hotspots are found to be associated with urban areas (these are discussed further in the manuscript). Still for others, no clear point-source could be identified. We classified all the discovered hotspots following these categories (Supplementary Table 1) and located them on a global map (Fig. 3). The latter reveals a high density of $C_2H_4$ hotspots in Europe, Russia, the Middle East, India, and East Asia (mostly in China and Japan), which together represent ~75% of the sites. China alone contains 113 hotspots. Note that a single $C_2H_4$ hotspot may originate from several closely located point-sources of different types. For instance, the vast Al Jubail

industrial complex (Saudi Arabia), shown in Fig. 2, includes petrochemical hubs and integrated steel plants. In such cases, only the two dominant categories at this location are displayed in Fig. 3 and Supplementary Table 1.

Overall, among the $C_2H_4$ hotspots, 138 (41%) are found to be associated with chemical industry, mostly in the Northern Hemisphere (Fig. 3). Many are part of a large petrochemical hub, such as the series of hotspots observed along the coast of Texas (United States) and Japan (Fig. 3), and around the Persian Gulf (Fig. 2). The production and processing of light olefins—ethylene and propylene—is usually at the center of every chemical and petrochemical complex. The leading route for olefin production is by steam cracking of crude oil or natural gas[19–21]. Al Jubail (Saudi Arabia) and Asalouyeh (Iran), highlighted in Fig. 2, are archetypes of petrochemical clusters with high-capacity ethylene crackers (1–1.5 Mt yr$^{-1}$). In such industries, $C_2H_4$ emission takes place from fugitive releases, gas flaring, and stack plumes resulting from the burning of fossil fuels[12,25,26]. In China and South Africa mainly, petrochemical point-sources located inland are often associated with coal-related activities in the same hotspot (Fig. 3). In such countries with limited oil and gas reserves, coal-to-olefins conversion technologies arise as an alternative to conventional steam cracking[20,21].

Overall, we identified coal exploitation and processing in 91 (27%) of the detected $C_2H_4$ hotspots. As a product of pyrolytic reactions within the flame, ethylene is emitted from the heating and burning of coal, at higher rates with decreased combustion efficiency. A vast majority of such hotspots associated with coal activities (76%) are located in China (Fig. 3). The industrial valley of the Shanxi province (Supplementary Fig. 4), where coal-related point-sources are observed in most hotspots in association with other heavy industries, is a typical example. Several of these point-sources are attributed to coal refining plants, graded coal (coke) being an important feedstock in sectors like petrochemistry and metallurgy. Other sources correspond to coal-fired power plants, especially in China where they still dominate the energy supply sector[27,28] and severely degrade air quality[29,30].

Point-sources related to metallurgy—mostly iron and steel plants—are identified in 85 (25%) of the $C_2H_4$ hotspots. India, China, Japan,

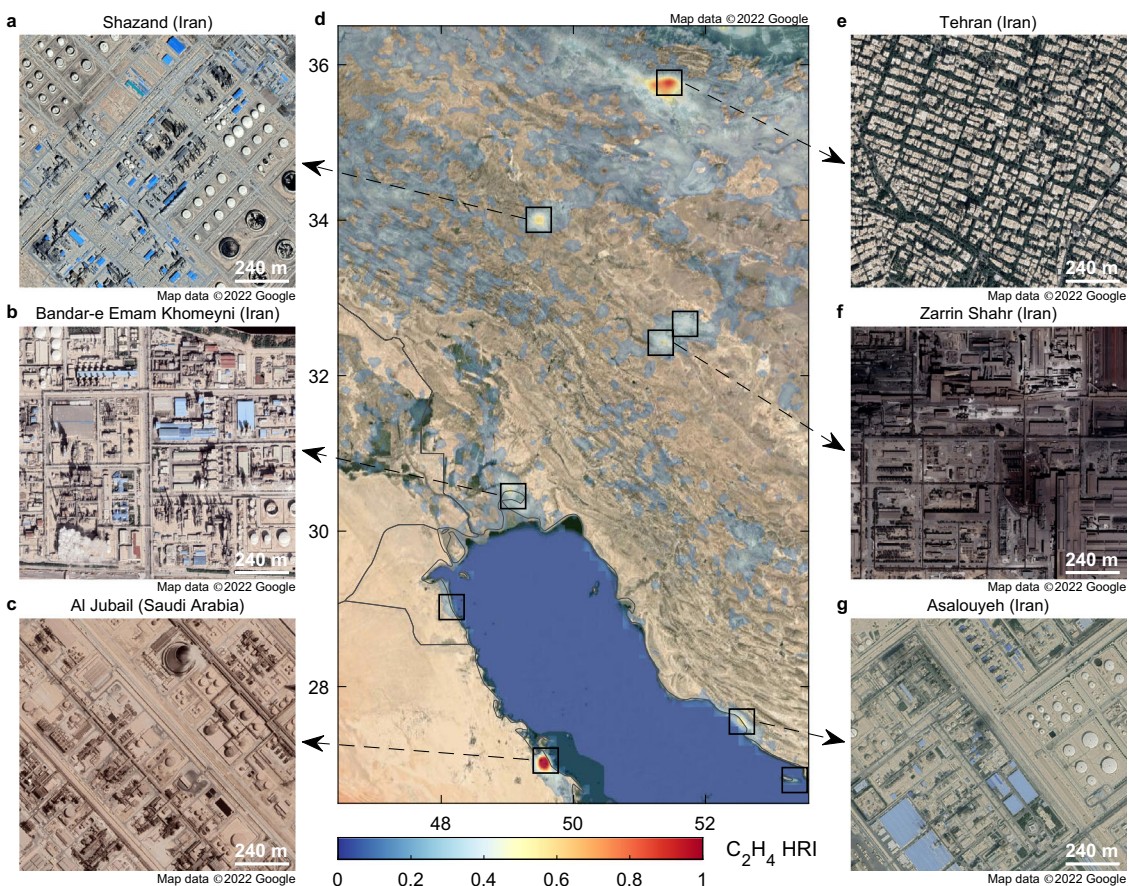

**Fig. 2 | IASI hyperfine resolution distribution, with hotspots and point-sources of ethylene. d** Zoom-in of the $C_2H_4$ HRI from the 13-year IASI average super-imposed on satellite visible imagery, over Iran and the Persian Gulf. HRI hyper-spectral range index; IASI Infrared Atmospheric Sounding Interferometer. Hotspots of ethylene are indicated with black squares. **a–c**, **e–g** Examples of close-up views on point-source emitters. Visible imagery from Google Earth, CNES/Airbus, Digi-talGlobe, and Landsat/Copernicus. Map data ©2022 Google.

Russia, and Ukraine concentrate most of these, with other noticeable presences in Mexico and Northern Africa (Fig. 3). Here, we differentiate metallurgy from the coal sector since the corresponding plants represent characteristic emitters that are easily identifiable on satellite visible imagery. Zarrin Shahr (Iran; Fig. 2) and Hejin (Shanxi, China; Supplementary Fig. 4) are typical examples of integrated steel plants. Those using coke obtained from bituminous coal are sometimes associated with local coal mining and processing (Fig. 3). Alternatively, iron and steel plants using petroleum coke (petcoke) derived from oil refining are also found in the proximity of petrochemical hubs (Fig. 3), such as in the Al Jubail industrial area (Fig. 2).

## Discussion

Constraints from satellite observations are crucial for evaluating state-of-the-art emission inventories such as EDGAR (Emissions Database for Global Atmospheric Research)[31], which in turn feed chemical models. Here, we provide spaceborne estimates of $C_2H_4$ fluxes for a suite of anthropogenic point-sources, which we compare with EDGAR v4.3.2. To this end, we converted the IASI HRI dataset to $C_2H_4$ total columns, and calculated the 13-year averaged fluxes from hotspots of interest via an inverse approach ("Methods"). The retrieval of $C_2H_4$ total columns being challenging, we limit our estimates to the hotspots with the highest HRI values and to those presenting the largest contrasts rela-tive to the surrounding background. In total, 57 global hotspots (53 related to industries and 4 to megacities) were quantified, repre-sentative of the different source categories and the global distribution of hotspots identified from space. Examples of IASI-based fluxes cal-culation are presented in Supplementary Figs. 6–8. Figure 4a depicts

the ratios between the IASI-based and EDGAR emissions for each of the 57 selected hotspots. This comparison indicates that EDGAR, with all sectors combined (i.e., transport, energy, residential, industries), underestimates the $C_2H_4$ fluxes, with ~50% of the IASI hotspots underpredicted by at least one order of magnitude. When only the emissions from the industrial sectors are considered, this percentage rises to ~75%, including ~38% underestimated by at least two orders of magnitude. We expect these biases to be even larger since the IASI-derived fluxes likely underestimate the real emissions because of the conservative $C_2H_4$ lifetime of 12 h assumed in the flux calculation ("Methods"), and of the reduced satellite sensitivity to the lowermost tropospheric layers.

Figure 4b, c give two examples of mismatch between IASI and EDGAR. In Fig. 4b, EDGAR predicts no or negligible industrial $C_2H_4$ fluxes over the IASI hotspot, which corresponds to the Dahej petro-chemical hub (Gujarat, India). This occurs similarly for 35 (66%) out of the 53 industrial hotspots studied here, suggesting that major indus-trial VOC hotspots are absent in bottom-up inventories. In the other example (Fig. 4c), $C_2H_4$ emissions over the industrial Fangshan District (Beijing, China) identified by IASI are accounted for, but under-predicted by EDGAR. Furthermore, the predicted fluxes are dominated in the inventory by diffuse releases from the transport and residential sectors, and do not reflect the presence of a strong industrial emitter. These discrepancies can be ascribed to the uncertainties on the total VOC emissions from existing databases, and to the difficulty, for inventories like EDGAR, to disaggregate the bulk of these emissions into spatially resolved fluxes of individual species. This is usually done by applying speciation profiles that are often fragmentary and

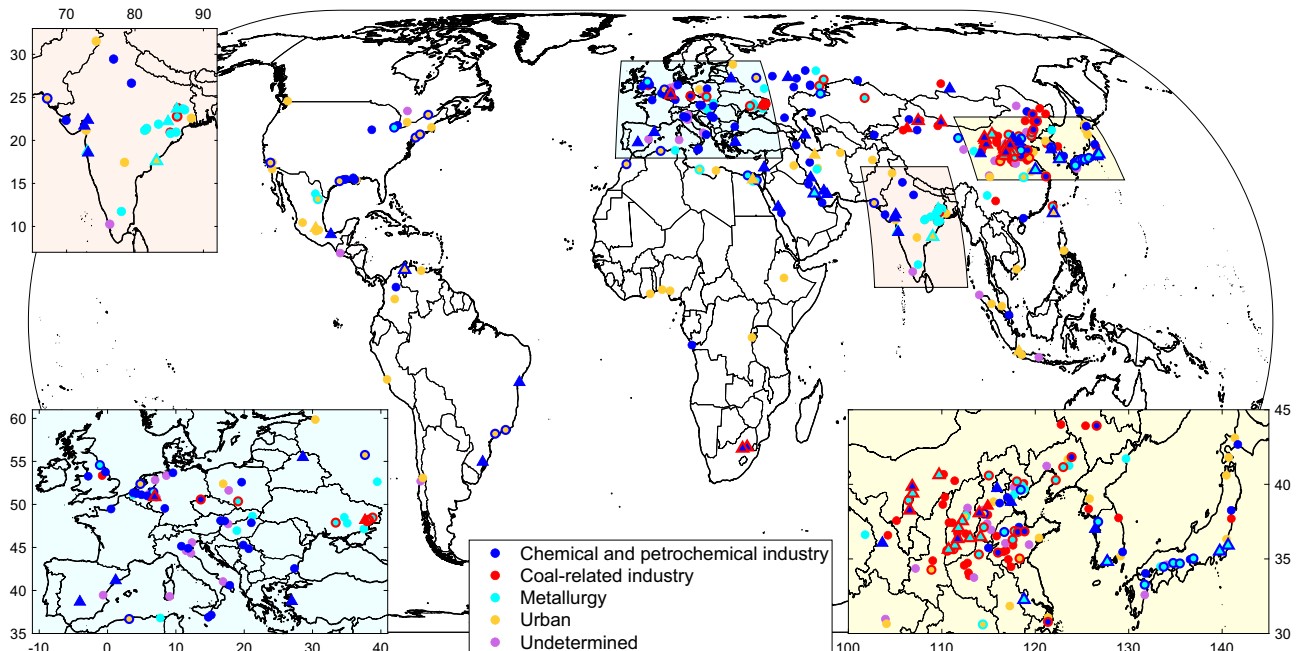

**Fig. 3 | Global distribution of $C_2H_4$ point-sources detected by IASI and their categorization.** A total of 336 hotspots were identified. When a hotspot belongs to more than one category, only the two main categories are depicted on the map. The $C_2H_4$ emission fluxes are calculated for 57 hotspots represented as triangles. IASI Infrared Atmospheric Sounding Interferometer.

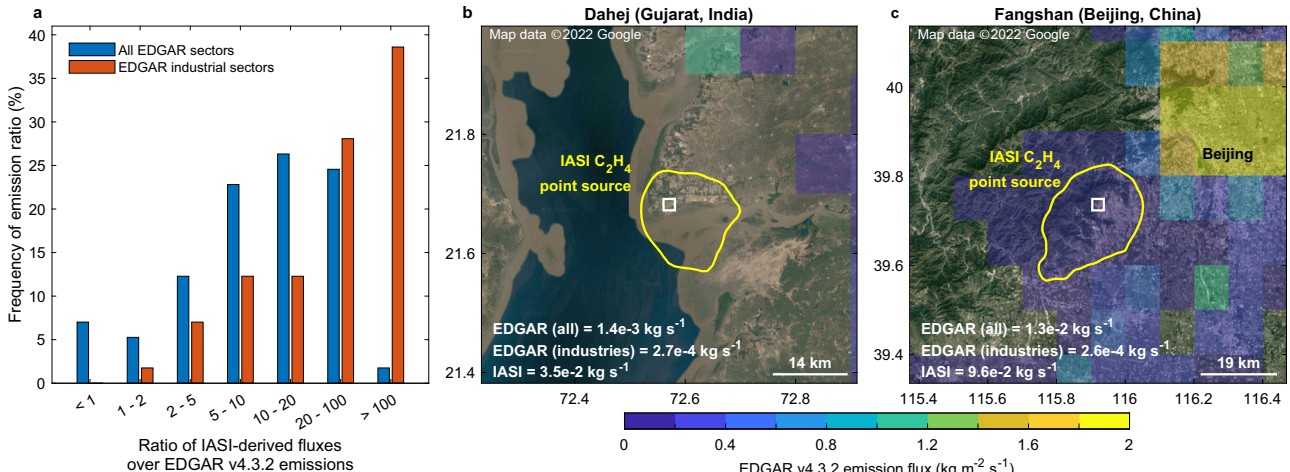

**Fig. 4 | Comparison between satellite-derived and EDGAR $C_2H_4$ fluxes. a** Ratio between the IASI-derived and EDGAR v4.3.2 $C_2H_4$ emissions, over 57 selected hotspots. IASI Infrared Atmospheric Sounding Interferometer; EDGAR Emission Database for Global Atmospheric Research. EDGAR emissions were computed, respectively, for all sectors (blue) and the industrial sectors only (red). The IASI-derived fluxes were calculated assuming a $C_2H_4$ lifetime of 12 h. The same figure, with a $C_2H_4$ lifetime of respectively 2 and 24 h, can be found in Supplementary Fig. 9. **b, c** Spatial mismatch examples between the EDGAR $C_2H_4$ fluxes (background color) and the IASI hotspot (yellow contour) delimited by the 90th percentile column value in the area. The white square indicates the location of the source emitter. Visible imagery from Google Earth, CNES/Airbus, DigitalGlobe, and Landsat/Copernicus. Map data ©2022 Google.

extrapolated to different geographical areas, emission sectors, and fuel types[31]. Consequently, numerous uncertainties add up in the bottom-up estimates of VOC emissions.

In contrast to industrial areas, EDGAR predicts overall large $C_2H_4$ releases from megacities, which are sometimes not captured by IASI, as illustrated with Beijing in Fig. 4c. Indeed, the satellite $C_2H_4$ emission values are lower than in EDGAR for 2 out of the 4 urban hotspots for which fluxes were calculated. In urban environments, traffic, residential heating, and wastes burning are the main contributors to the elevated ambient $C_2H_4$ concentrations[9,32,33]. As these releases are more diffuse than in industrial areas, emitted ethylene is more difficult to

detect from space. Conversely, the industrial emissions, being more concentrated and often associated with the presence of high stacks, are more prone to be uplifted at altitudes where the detection by IASI is easier. Although generally weaker than the industrial enhancements, still 77 (23%) hotspots captured from space are linked to megacities, among which the most noticeable are Tehran (Fig. 2), Mexico City, Cairo, Jakarta, Kuala Lumpur, and Saigon. The improved sensitivity of future satellite sounders to the lower tropospheric layers will help to track down further urban point-sources. In particular, the forthcoming IASI-New Generation (IASI-NG), with double the radiometric and spectral resolution performances of its predecessor[34], will offer

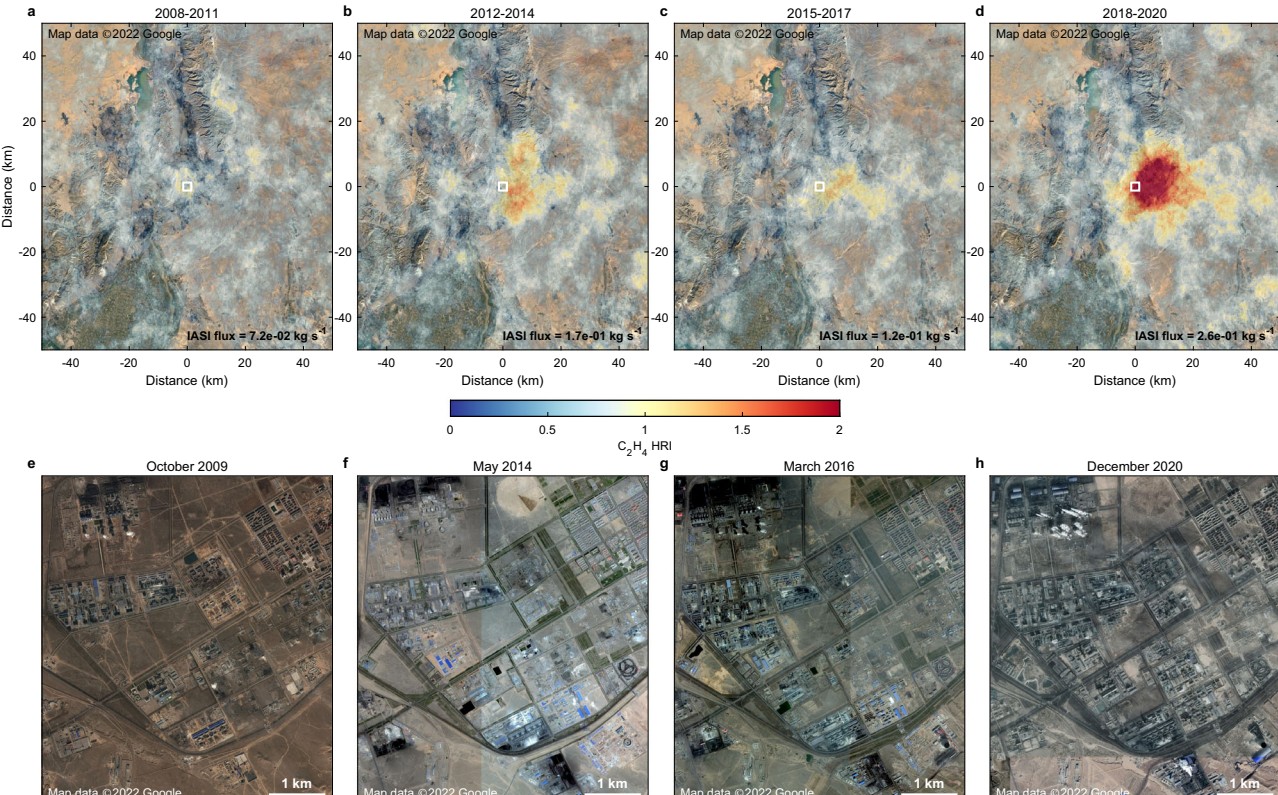

**Fig. 5 | Temporal assessment of a $C_2H_4$ point-source in China. a–d** Wind-rotated supersampling of the IASI $C_2H_4$ HRI at a 0.01° × 0.01° spatial resolution, around the hotspot of the Hainan District (Inner Mongolia, China), over 2008–2011, 2012–2014, 2015–2017, and 2018–2020. HRI hyperspectral range index; IASI Infrared atmospheric Sounding Interferometer. The white squares indicate the location of the source emitter(s). The top-down $C_2H_4$ emission fluxes calculated for each time period are provided in black. **e–h** Zoom-ins on the presumed emitter(s) with satellite visible imagery taken within each time period. Visible imagery from Google Earth, CNES/Airbus, DigitalGlobe, and Landsat/Copernicus. Map data ©2022 Google.

enhanced measurement capacity of hotspots of ethylene, and of the VOCs in general.

To investigate whether IASI can reveal temporal changes in the anthropogenic emissions of atmospheric $C_2H_4$, we applied the wind-adjusted super-resolution technique to the satellite data over four different time periods: 2008–2011, 2012–2014, 2015–2017, and 2018–2020. Despite the decrease in the signal-to-noise ratio, the representation of the point-sources and transport plumes is good enough to allow studying the time evolution of the largest hotspots. For most, no significant trends were observed, but for others, the hyperfine resolution maps reveal a clear progressive enhancement of the downwind $C_2H_4$ HRI average with time, as illustrated in Fig. 5 for a coal-related hotspot in Inner Mongolia, China. Another example over a large petrochemical hub in Saudi Arabia is presented in Supplementary Fig. 10. For these two examples, the top-down $C_2H_4$ emissions grow respectively from $7.2 \times 10^{-2}$ and $8.8 \times 10^{-2}$ kg s$^{-1}$ for the 2008–2011 period, to $2.6 \times 10^{-1}$ and $2.9 \times 10^{-1}$ kg s$^{-1}$ over 2018–2020. Satellite visible imagery supports this finding, as new industrial complexes are seen to appear over the different periods (Fig. 5, Supplementary Fig. 10). Similar examples are found in the central and western regions of China, which have recently undergone programs of industrial development that rely heavily on the exploitation of local coal resources as raw materials and for energy production[27,28]. Currently, the temporal assessment is limited to the most prominent point-sources and to multiyear time blocks. These examples, however, demonstrate that infrared sounders can be used to monitor industrial emissions of VOCs, a capability that is surely going to improve with the next-generation infrared sounders that will offer better spectral resolution and lower noise in case of IASI-NG[34], and increased temporal sampling in case of the geostationary infrared sounder onboard the Meteosat

Third Generation (MTG-IRS) satellite (https://www.eumetsat.int/meteosat-third-generation).

The identification and attribution of anthropogenic point-sources remain challenging, notably due to the numerous emission and formation processes of the usual short-lived pollutants. Especially, while $NO_2$ and $SO_2$ indicate mainly the presence of combustion and smelting activities[1,2], and $NH_3$ originates primarily from agriculture and industrial fixation of nitrogen[3,24], none of them point directly to human-related emissions of VOCs at the origin of important air pollution[5–9]. In this respect, our results demonstrate that ethylene complements these inorganic pollutants as a key short-lived carbon tracer for tracking down and identifying point-sources specific to heavy industries.

## Methods

### $C_2H_4$ retrieval and IASI dataset
The measurements of ethylene ($C_2H_4$) are derived from the hyperspectral observations recorded by the Infrared Atmospheric Sounding Interferometer (IASI), flying on the three sun-synchronous, polar-orbiting meteorological satellite platforms Metop[22]. IASI/Metop-A, -B, and -C are providing data since, respectively, October 2007, March 2013, and September 2019 (IASI/Metop-A was decommissioned in late 2021). In this work, the three IASI-A, -B, and -C $C_2H_4$ datasets are used together and show an excellent agreement during their overlapping periods. IASI is a Fourier transform spectrometer with an apodized spectral resolution of 0.5 cm$^{-1}$ (spectrally sampled at 0.25 cm$^{-1}$), which measures in a nadir geometry the radiance of the Earth and of the atmosphere in the thermal infrared spectral range between 645 and 2760 cm$^{-1}$ without gap[22]. The radiometric noise in the spectral range of the main $C_2H_4$ absorption feature near 949 cm$^{-1}$ is ~0.15 K for a reference blackbody at 280 K. One IASI instrument provides near global

coverage twice a day, with measurements at -09:30 am and pm (local equator crossing time). At nadir, the footprint of an IASI measurement is a 12 km diameter circle, and at off-nadir angles, an ellipse elongated up to $20 \times 39$ km.

The detection and retrieval of ethylene were performed with version 3 of the Artificial Neural Network for IASI (ANNI) v3. This versatile scheme has been developed specifically to allow a robust retrieval of weakly absorbing trace gases from the hyperspectral IASI observations and is now used extensively for the retrieval of ammonia ($NH_3$)[35,36], volatile organic compounds (VOCs)[37–40], and dust[41]. Therefore, we provide here only a summary of the method and all the elements that are specific to the retrieval of ethylene, and we refer to the above-mentioned papers and other references therein for a detailed description of the ANNI procedure.

The ANNI method proceeds in two main steps. First, the detection of the target gas is made in each IASI spectrum via the calculation of a HRI, a dimensionless metric of the strength of the signature of a target absorber in the recorded spectrum[42]. For each species, the HRI is set up over a spectral range that typically comprises all the absorption features of the target compound, while avoiding the ranges with major interferences or limited information content. This detection method is proven to be very sensitive and well-suited for the detection of the VOCs[37–40], which are characterized by weak and broadband absorptions in the thermal infrared[35–41]. Subsequently, for each IASI measurement, the corresponding HRI is converted in a single-pixel gas total column (with an associated uncertainty) with the help of an artificial feedforward neural network (NN). Such NN emulates a close approximation of the complex system that binds together the HRI, the gas abundance, and the state of the Earth's atmosphere and surface, with a high computing efficiency. A NN is trained specifically for each target species from an extensive synthetic training set based on real IASI observations and built in such a way that it encompasses all the possible conditions of atmosphere and gas abundance. The trained NN is tested carefully to ensure that it generalizes well, i.e., that it is able to provide realistic results for all abundances of the target species and states of the atmosphere encountered in real observations.

The $C_2H_4$ HRI is calculated over the $940–960$ cm$^{-1}$ spectral range, which comprises the main $C_2H_4$ absorption feature in the thermal infrared (its Q-branch of the v7 vibrational band). By construction[42], the HRI is normalized, so that its statistical distribution is a Gaussian with a mean of zero and a standard deviation of 1 when calculated on background spectra (i.e., without observable amount of $C_2H_4$). For an individual observation, one can therefore consider that a clear detection of $C_2H_4$ is achieved for an HRI value of at least 3. However, it is worth noting that, in the maps presented in this study, the detection threshold is lowered considerably owing to the large number of satellite measurements averaged per grid cell. Further, we present an unambiguous detection of ethylene in IASI spectra over $C_2H_4$ enhancements (see also Supplementary Fig. 5).

For the NN setup, we followed closely the ANNI v3 procedure applied to the $NH_3$ retrieval[35,37,43]. The NN was built via a training phase based on an extensive dataset made of all the necessary input and output variables. The variables feeding the NN are the HRI, the temperature profile, an $H_2O$ profile, the surface pressure and emissivity, a spectral baseline temperature, and the IASI viewing angle. The outputs consist of the gas total column and an uncertainty on the retrieved column[35,37]. The training set was built from the simulation of -500,000 IASI spectra performed with a line-by-line radiative transfer model, ensuring a homogeneous representation of a large range of observational conditions and gas abundance. Specifically, the simulations encompass thermal contrasts from $-30$ to $40$ K, and $C_2H_4$ total columns from $<1 \times 10^{14}$ to $-1 \times 10^{17}$ molecules cm$^{-2}$. The thermal contrast is defined as the temperature difference between Earth's surface and overlying air layer. To convert the HRI in gas column in the forward simulations, a vertical profile of ethylene was parameterized with a

Gaussian function, similar to what was done for ammonia[35,36]. Here, the volume mixing ratio of ethylene $vmr_{(C_2H_4)}$ at an altitude $z$ (in km) is defined as:

$$vmr_{(C_2H_4)} = k \times e^{-\frac{(z-z_0)^2}{2\sigma^2}} \tag{1}$$

with $z_0$ the peak height of the vertical profile (in km) as a measure of the altitude of the bulk of $C_2H_4$, $\sigma$ the standard deviation of the Gaussian function as a measure of the thickness of the $C_2H_4$ layer (in km; assigned as explained below), and $k$ (in ppb) a scaling factor of the profile that controls $vmr_{(C_2H_4)}$ at $z_0$ and hence the $C_2H_4$ abundance in the forward simulations. In this work, the peak concentration was always fixed at the surface to be representative for observations over hotspots, as the bulk of $C_2H_4$ is assumed to be close to the surface. The NN trained for ethylene is made of two computational layers of 12 nodes; a satisfactory performance similar to that of $NH_3$[36] and other VOCs[37–40] was indeed reached with this network architecture, while keeping the NN relatively small to prevent overfitting.

During the retrieval process, the meteorological input variables feeding the NN were taken from the European Center for Medium-Range Weather Forecasts (ECMWF) ERA5 reanalysis[44], which guarantees a full consistency throughout the IASI operational time series[43]. Available on an hourly timescale with a 0.28125° resolution, the ERA5 data were collocated with the IASI measurements. The value of $\sigma$ was set to the ERA5 boundary layer height. Consistent with the other products of the ANNI framework, a pre-filter prevents the retrieval on cloudy scenes, and a post-filter discards the $C_2H_4$ columns affected by too large uncertainties or by poor observational conditions to retrieve ethylene. In practice, the retrieved column is rejected when the ratio $|column_{(C_2H_4)}/HRI_{(C_2H_4)}|$ is higher than $5 \times 10^{16}$ molecules cm$^{-2}$ or when the spectral baseline temperature is lower than $265$ K. It was shown that this type of post-filter, based on the column-to-HRI ratio, is primarily driven by the thermal contrast of the observation scene, i.e., by the surface and atmospheric conditions, and not directly by the gas abundance[35,39]. Typically, it allows removing unphysical retrievals when poor observational conditions limit the ability of IASI to detect the target species.

An uncertainty on each retrieved column is evaluated by propagating the uncertainties of the different input variables of the NN, as detailed by refs. 35,37,39. The typical uncertainty on an individual retrieved $C_2H_4$ column is below 50% for columns above $1 \times 10^{16}$ molecules cm$^{-2}$ and positive surface-atmosphere thermal contrasts. Consistent with the other VOCs retrieved from IASI[37–39], the error values increase beyond 50% for lower columns as the weak $C_2H_4$ concentrations approach the IASI detection threshold, and for weak or negative thermal contrasts which reduce the IASI sensitivity. However, these uncertainties are significantly reduced for the column averages calculated here, due to the large number of measurements per grid cell (see below).

To evaluate the error on the IASI vertical profile, we used (Eq. 1) to build two $C_2H_4$ vmr profiles assuming a peak concentration of 5 ppb ($vmr_{(C_2H_4)}$ at $z_0$) at, respectively, surface and 0.5 km altitude. Using the 1976 US standard atmosphere, we calculated $C_2H_4$ total columns of $2.13 \times 10^{16}$ and $2.64 \times 10^{16}$ molecules cm$^{-2}$ from these two profiles. This 20% difference is well below the typical 1–2 order(s) of magnitude of discrepancy that we observe between the top-down and EDGAR emissions over industrial point-sources.

In this work, the satellite dataset that is exploited, for both the search of $C_2H_4$ HRI enhancements and the calculation of $C_2H_4$ fluxes over hotspots, consists of the 2008–2019 IASI/Metop-A, the 2013–2020 IASI/Metop-B, and the 2020 IASI/Metop-C observations taken over land. Only the satellite data from the morning overpasses are exploited to take advantage of the overall larger thermal contrast during daytime, which translates to a larger sensitivity of the IASI

measurements to the lower tropospheric layers. In addition, the observations affected by >10% cloud coverage are discarded from the IASI dataset. Finally, we exclude all the data outside the 60° S–70° N latitudinal band, as anthropogenic gas emissions are limited in polar regions. Furthermore, these high latitudes are largely affected by cold temperatures and weak thermal contrasts, and hence offer to the satellite sounders poor measurement sensitivity to the trace gases. After applying the different filters, over $1.4 \times 10^9$ IASI measurements are here exploited.

A preliminary evaluation of the IASI retrievals is presented in Supplementary Table 3. It compares the satellite data over a suite of hotspots with independent columns derived from (near-)surface vmr data found in the literature and measured in the vicinity of these hotspots. The vmr data were converted in total columns by assuming the same vertical $C_2H_4$ distribution as the Gaussian function implemented by ANNI v3 (Eq. 1), scaled at the surface to match the vmr data. The 1976 US standard atmosphere and a planetary boundary layer height ($\sigma$) at 1 km altitude are assumed to keep this analysis straightforward and synthetic. The comparison shows that the IASI $C_2H_4$ columns fit well within the ranges of abundance derived from the independent measurements and gives confidence in the satellite retrievals over the hotspots.

## Oversampling and super-resolution techniques

Oversampling techniques allow increasing the spatial resolution of satellite data beyond the native resolution of the sounder measurements (i.e., pixels of 12 km diameter at nadir for IASI). It is achieved by combining many satellite samplings of the same scene and by exploiting the varying footprint on the ground of the satellite pixels, which partially overlap after successive satellite overpasses[3,45]. Where measurement footprints intersect, sub-pixel information becomes available and allows to be used to create high-resolution maps. In those, the oversampling reveals features of small geographical extent, such as hotspots of a short-lived gas tracer, which would not be seen or would be hardly detectable by regular binned averaging of satellite data[1–3]. This is illustrated for a $C_2H_4$ hotspot (Mengxi Park, Inner Mongolia, China) in Supplementary Fig. 1a, b.

As oversampling usually requires hundreds of satellite measurements over the same area, we took advantage of the relatively long observational time series of IASI, and of the three instruments in operation, to assemble an extensive dataset of over $1.4 \times 10^9$ spectra to track down $C_2H_4$ hotspots. The supersampling technique applied here is an augmented oversampling procedure developed for an advanced search of $NH_3$ hotspots with IASI[24]. It combines two concepts: the wind rotation and the supersampling.

The application of the wind rotation to short-lived trace gases is described by refs. 46,47. It consists in rotating each satellite measurement around the presumed gas emitter according to the daily horizontal wind direction. Applied to a satellite time series, this yields a distribution of the measurements in which the winds blow in the same direction from the point-source. The benefit of this technique combined with the oversampling is presented in Supplementary Fig. 1c, which depicts the oversampled distribution of the $C_2H_4$ HRI obtained by applying beforehand the daily wind rotation of the IASI data around a point-source. Hereafter, we refer to this high-resolution distribution as the downwind average. Specifically, as the winds are aligned in the same direction (to the east in Fig. 1c), the resulting gas distribution exhibits hotspots with enhanced magnitude and more concentrated transport plumes of the pollutant from the presumed emitter. It also reduces the contribution of nearby sources. Here, the wind rotation is based on the daily horizontal wind fields from the ECMWF ERA5 reanalysis[44].

Described by ref. 24, the supersampling (or super-resolved oversampling) is derived from super-resolution techniques as it aims at reconstructing high-resolution images from numerous low-resolution representations of the same scene. It consists in repeating the oversampling procedure and in correcting, at each iteration, the resulting oversampled average according to the differences with the ground truth (the satellite observations). Specifically, the solution of the supersampling for the first iteration corresponds to the oversampling. Then, we calculate the differences between the single-pixel IASI measurements and the same individual observations simulated by assuming that this first-iteration supersampling is the ground truth. For the second iteration, the solution consists in adding the oversampled average of these measurement differences to the supersampling solution from the previous step. This procedure is repeated until a supersampled average that is optimally consistent with the ground truth is found. A condition for the application of this technique is to assume that the underlying satellite data distribution is relatively constant in time. As demonstrated by ref. 24, redistributing the daily IASI measurements according to the wind fields, as done with the wind rotation presented above, allows removing most of the variability over a gas point-source and guarantees the required homogeneity. Supplementary Figs. 1–2d show examples of downwind average of $C_2H_4$ HRI obtained by applying the supersampling to the daily wind-rotated IASI maps around a point-source. While the oversampling typically smooths out the satellite data, the supersampling reproduces more realistically the strength of the hotspots and resolves much finer spatial features (of about 3–4 km with IASI). This allows the discovery of hotspots that would be difficult to find with oversampling alone, as illustrated in Supplementary Fig. 2. For instance, it allowed doubling the number of $NH_3$ point-sources identified with IASI[24] compared to those detected with oversampling[3]. It is worth noting that the super-resolution preserves the gas mass around the point-source with respect to the original IASI data, as shown by ref. 24.

To track down the $C_2H_4$ hotspots, we have applied the wind-rotated supersampling to the IASI dataset of $C_2H_4$ HRI following the procedure described by ref. 24. Specifically, each location on Earth is treated as a potential point-source, and the wind-rotated supersampling is applied successively to each grid cell of a 0.01° × 0.01° world map. Supplementary Fig. 1d illustrates the application of this technique to one location. In this example, Mengxi Park (Inner Mongolia, China) is the presumed emitter and is used as the wind rotation center. In the distribution resulting from the supersampling, an averaged HRI value is calculated over the area downwind of the emitter. This value is then attributed to the 0.01° × 0.01° grid cell that corresponds to this point-source on the world map. The procedure described here is repeated independently for each grid cell. The result is a global $C_2H_4$ distribution at a hyperfine 0.01° × 0.01° spatial resolution in which, to each grid cell, has been assigned the HRI value of the downwind average obtained when this grid cell is assumed to be an emitter (Fig. 1). Zoom-ins of this distribution on regions with a high concentration in $C_2H_4$ hotspots are presented in Fig. 2 and Supplementary Fig. 4. A thorough visual analysis of this distribution allowed the detection of 336 global hotspots (Fig. 3; Supplementary Table 1). These correspond typically to areas of 20–50 km spatial extent with HRI values significantly higher than the surrounding background. Additional satellite visible imagery and third-party sources were used to exclude false detections due to e.g., fires or emissivity features (see further).

The reason why we searched for the $C_2H_4$ hotspots with the HRI dataset, and not directly with the columns, is that the HRI is correlated to the gas abundance, without being affected by the uncertainties associated with the retrieved column. Consequently, as illustrated in Supplementary Fig. 3, the resulting $C_2H_4$ distribution is less noisy and offers a better representation of the hotspots.

## Whitening transformation of IASI spectra

Following the ANNI v3 procedure, ethylene is detected and quantified in an individual IASI spectrum by means of the HRI, which is calculated

over a specific spectral range. Following ref. 42, the HRI is defined as:

$$\text{HRI} = \frac{K^T S_y^{-1}(y - \bar{y})}{\sqrt{K^T S_y^{-1} K}} \frac{1}{N} \qquad (2)$$

$K$ is the $C_2H_4$ spectral Jacobian (i.e., its spectral signature), $\bar{y}$ a mean background spectrum calculated from a set of observations representative for background conditions (i.e., without an observable amount of $C_2H_4$), $S_y$ the covariance matrix associated with the calculation of $\bar{y}$, and $N$ a normalization factor. By definition, the HRI distribution has a zero mean and a standard deviation of 1 when calculated on IASI measurements that do not contain detectable $C_2H_4$. Although being very sensitive to the detection of weak absorbers, this metric can be prone to false detections when there is a partial match between the $C_2H_4$ spectral signature and the one of an interference (e.g., another trace gas or surface emissivity artefacts)[41]. Therefore, a firm identification of ethylene is needed to confirm its significant contribution to the HRI enhancements pinpointed as $C_2H_4$ hotspots.

Recently, a transformation of the spectral channels, referred to as whitening, has been applied for the first time to the IASI spectra and allowed the identification of several trace gases that have never been detected before with nadir satellite sounders[48]. The whitening is similar to the HRI concept in that it removes most of the climatological background of the analyzed spectra (Eq. 3), resulting in a spectrum $\tilde{y}$ in which each channel has been transformed in a normalized, uncorrelated variable:

$$\tilde{y} = S_y^{-1/2}(y - \bar{y}) \qquad (3)$$

The main difference between the HRI and the whitening is that this latter is not a sum over all the channels and is not specific to a particular species (i.e., $K$ does not appear in Eq. 3). Therefore, while the HRI provides a unique value, the result of the whitening transformation (Eq. 3) is a spectrum $\tilde{y}$ in which all the spectral residuals that differ from the climatological background are exposed. Spectral anomalies can be assigned to specific trace gases, by comparing $\tilde{y}$ with the whitened Jacobian $\tilde{K}$ of candidate absorbers[48], which is defined as:

$$\tilde{K} = S_y^{-1/2} K \qquad (4)$$

A match between a spectral anomaly and the $\tilde{K}$ of a species helps to demonstrate the presence of an enhanced amount of this compound in the analyzed spectra. Note that using the previous definitions (Eqs. 2–4), both concepts of HRI and whitening are related as follows:

$$\text{HRI} = \frac{\tilde{K}\tilde{y}}{N} = \frac{\tilde{K}^T \tilde{y}}{|\tilde{K}|N} = \sum_{i=1}^{n} \text{HRI}_i = \sum_{i=1}^{n} \frac{\tilde{K}_i \tilde{y}_i}{|\tilde{K}|N} \qquad (5)$$

The HRI of a target species, which is integrated over a whole spectral range, is indeed the sum of all the contributions $\text{HRI}_i$ from each channel i within that range and is also the product of the whitened spectrum $\tilde{y}$ with the whitened Jacobian $\tilde{K}$ of that species.

For a suite of $C_2H_4$ hotspots, we have applied the whitening transformation to the average of spectra taken within a radius of 20 km around the hotspot. Examples are presented in Supplementary Fig. 5a–d. For each of them, the top panel shows the whitened mean spectrum $\tilde{y}$, which displays the spectral anomalies compared to the background, and the whitened Jacobian $\tilde{K}$ of ethylene. The wavenumbers that are displayed (940–960 cm$^{-1}$) correspond to the range used to calculate the $C_2H_4$ HRI. In all the examples, we observe spectral anomalies around 949.5 cm$^{-1}$, corresponding to the position of the $C_2H_4$ $Q$-branch. This already represents a clear identification of ethylene and a confirmation of its absorption signature in the IASI spectra.

Most other anomalies are attributed to water vapor and, to a lesser extent, to ammonia.

The bottom panel shows the product of the whitening residuals with the whitened spectral signature of $C_2H_4$, which allows us to visualize the contribution of every channel to the HRI. The resulting curve indicates clearly that the channels around 949.5 cm$^{-1}$ are by far the largest contributors to the total HRI. This becomes obvious when calculating three partial HRI's over successively the 940–948.25 cm$^{-1}$, 948.5–951 cm$^{-1}$, and 951.25–960 cm$^{-1}$ ranges, denoted hereafter $\text{HRI}_{r1-r3}$. Indeed, based on Eq. 5, one can write:

$$\text{HRI} = \text{HRI}_{r1} + \text{HRI}_{r2} + \text{HRI}_{r3} = \frac{\tilde{K}_{r1}^T \tilde{y}_{r1}}{|\tilde{K}|N} + \frac{\tilde{K}_{r2}^T \tilde{y}_{r2}}{|\tilde{K}|N} + \frac{\tilde{K}_{r3}^T \tilde{y}_{r3}}{|\tilde{K}|N} \qquad (6)$$

In all the examples, the value of $\text{HRI}_{(r2)}$, the narrow window encompassing almost exclusively the $C_2H_4$ $Q$-branch, is the highest; this constitutes strong spectral evidence that ethylene is the main contributor to the HRI enhancements detected by IASI over the hotspots. The non-negligible values of $\text{HRI}_{r1}$ and $\text{HRI}_{r3}$ are attributed to weaker $C_2H_4$ absorptions also present in these ranges, and to a partial match between the $C_2H_4$ signature and interferences (e.g., water vapor). It is worth mentioning that the spectral residuals assigned to such interferences are usually weighted to values close to zero by the weak $C_2H_4$ spectral signature in these channels. This is the case, for instance, for the two main $NH_3$ absorption features present at 948 and 951.75 cm$^{-1}$, i.e., outside the $\text{HRI}_{r2}$ window. As a result, ammonia can be considered here as a minor interference.

In the IASI hyperfine resolution map of ethylene (Fig. 1), we notice that the HRI background shows important spatial variations. While it is expected to obtain values close to zero over remote areas (e.g., deserts) due to the absence of local sources, sometimes high HRI values are observed over entire regions. Since some of these only contain few human activities, it is therefore important to understand whether ethylene contributes significantly to the observed regional enhancements or those result from false detections. Similar to the hotspots, we have applied the whitening transformation to averaged IASI spectra selected in a radius of 65 km over areas with higher background HRI values. Examples are presented as Supplementary Fig. 5e–h. For some regions, we reach the same conclusion as for the hotspots, that is, ethylene is the dominant contributor to the HRI enhancements observed in these areas. This can be assigned to regular biomass burning events that occur in the regions, as in Myanmar (Supplementary Fig. 5e), or to the elevated level of atmospheric pollution due to the presence of many anthropogenic sources, such as in northern France (Supplementary Fig. 5f). In other regions that are remote, ethylene is not the main reason for the high HRI values, yet its contribution might remain important, as in the Volgograd Oblast (Russia; Supplementary Fig. 5g), where vegetation and agricultural fires are relatively frequent. In the whitened mean spectrum of these regions, we have indeed identified a broadband residual in the 958–960 cm$^{-1}$ range caused by surface emissivity features (Supplementary Fig. 5g, h). Although the emissivity effect for one channel is dampened by lesser weight given to this range by the $C_2H_4$ Jacobian, the contribution of every channel that is affected adds up and increases the HRI value, leading to partially false detections over these regions.

## $C_2H_4$ emission flux calculations

For the most prominent hotspots identified with IASI (Supplementary Table 2), we obtained estimates of the $C_2H_4$ emission fluxes $E$ from the IASI column distribution by following an approach similar to the calculation of the $NH_3$ fluxes[3]. Examples are displayed in Supplementary Figs. 6–8. As described by ref. 3, we used the box model $E = M/\tau$, with $M$ the $C_2H_4$ total mass contained in the box, and $\tau$ the $C_2H_4$ effective lifetime. With this model, we assumed a steady

state and first-order loss terms, and we disregarded possible $C_2H_4$ transport out of the box. For each hotspot, the size of the box was adjusted in such a way to encompass most of the plume and minimize the amount of ethylene transported out of the area. $M$ was calculated directly from the $0.01° \times 0.01°$ distribution of $C_2H_4$ IASI columns produced by the wind-rotated supersampling (Supplementary Figs. 6–8a), using the coordinates of the $C_2H_4$ point-source (Supplementary Table 1) as the rotation point. The wind rotation preserves the distance of the satellite data to the presumed point-source, while concentrating the bulk of trace gas by aligning the wind fields. To account for an ambient level of ethylene in the area, we subtracted from the $C_2H_4$ column distribution a background, which was determined as the column averaged over a side band of the box that is not affected by the transported plume (Supplementary Figs. 6–8a). We obtained $M$ from the resulting column distribution (Supplementary Figs. 6–8b) by summing up the $C_2H_4$ masses (i.e., IASI column × surface area) in each $0.01° \times 0.01°$ grid cell. The global chemical lifetime of ethylene against reaction with OH and $O_3$, its main sinks in the atmosphere, is estimated to 1–1.4 day for averaged OH and $O_3$ concentrations in the range of $10^6$ and $10^{11}$ molecules $cm^{-3}$, respectively[5,49,50]. However, based on the same reaction rate coefficients, $C_2H_4$ lifetimes of a few hours are derived from field campaigns in industrial and urban areas due to the higher ambient concentrations in atmospheric oxidants ([OH] = ~$10^7$ molecules $cm^{-3}$)[6,8,51,52]. Therefore, we assumed a lifetime $\tau$ of 12 h to calculate the $C_2H_4$ emission fluxes from the point-sources. Considering that this lifetime is conservative for the highly polluted environments that are the $C_2H_4$ hotspots, the emissions obtained here likely underestimate the real fluxes. To account for the uncertainties on the $C_2H_4$ lifetime, we performed the same flux calculation assuming successively $\tau$ = 24 h and $\tau$ = 2 h (Supplementary Fig. 9; Supplementary Table 2).

Assuming a constant lifetime is a simplification in the calculation of the top-down fluxes. To account for the spatial and temporal variability of the atmospheric $C_2H_4$ lifetime, a step forward would consist in estimating this lifetime for each point-source, based directly on the satellite measurements themselves, as done for $SO_2$[47,53] and for $NO_2$[1]. However, this approach is currently not workable for $C_2H_4$, as it requires satellite measurements with very low noise, over both source and remote regions. With the future launch of IASI-NG, which will have much higher instrumental performance compared to its predecessor[34], such an approach will be more likely attainable.

For the same point-sources, we also calculated the $C_2H_4$ emission fluxes prescribed by the state-of-the-art EDGAR v4.3.2 anthropogenic inventory[31], using the data for the most recent available years (2010–2012). The EDGAR fluxes are provided at a spatial resolution of $0.1° \times 0.1°$ for a suite of VOC species and for different sectors. As described by ref. 31, speciation profiles are applied to disaggregate the total VOC emissions available in existing databases (e.g., national emission inventories) into sector-specific fluxes of individual VOCs, including ethylene, or of similar VOCs lumped together. Due to the lack of speciation profiles representative for all the emission sectors, types of fuels, and geographical areas, this disaggregation might be the source of biases in the predicted fluxes, cumulated to the large uncertainties associated with the total VOC emissions on which the speciation profiles are applied. For each of the selected hotspots, we computed the $C_2H_4$ emission fluxes by summing up the contribution of the EDGAR $0.1° \times 0.1°$ pixels located over and in the direct vicinity of the presumed point-source (Supplementary Figs. 6–8c, d). As many of these hotspots are relatively isolated, the contribution of the pixels around the point-source is usually small. However, especially in the case of a hotspot located over a large industrial or urban area, it cannot be ruled out that other sources nearby the presumed emitter also contribute to the $C_2H_4$

enhancement observed with IASI. This is accounted for in the flux calculation by also considering the pixels around the point-source. The EDGAR emission fluxes of ethylene are available for 16 sectors belonging to industrial, transport, and residential categories[31]. As most of the $C_2H_4$ hotspots identified with IASI are associated with industrial activities, we computed the EDGAR fluxes over successively all the sectors and only the sectors related to the industry. Those include power industry, oil refineries, transformation industry, combustion for manufacturing, fuel exploitation, and process emissions during production and application.

## Data availability
The super-sampled IASI $0.01° \times 0.01°$ $C_2H_4$ HRI dataset generated and analyzed in this study, the catalog of the identified and categorized $C_2H_4$ point-sources, the super-sampled IASI $C_2H_4$ total columns used to calculate the emission fluxes, and the source data, have been deposited in the Zenodo database (https://doi.org/10.5281/zenodo.7085725). The $C_2H_4$ emission fluxes from the EDGAR v4.3.2 inventory analyzed in this study are available on the EDGAR website at https://edgar.jrc.ec.europa.eu/dataset_ap432_VOC_spec#p3.

## Code availability
The code to calculate the $C_2H_4$ HRI from IASI spectra and to retrieve the $C_2H_4$ total columns, including the artificial neural network used for the retrievals, instructions, and example data, have been deposited in the Zenodo database (https://doi.org/10.5281/zenodo.7085725). The codes of the oversampling, wind rotation, and supersampling used in this work are available in the paper of Clarisse et al. (2019) at https://doi.org/10.5194/amt-12-5457-2019.

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

## Acknowledgements

The research has been supported by the HIRS Prodex arrangement (ESA-BELSPO). L.C. is research associate supported by the F.R.S.-FNRS. IASI is a joint mission of Eumetsat and the Center National d'Etudes Spatiales (CNES, France). The IASI Level-1C data are distributed in near real-time by Eumetsat through the EumetCast distribution system. The authors acknowledge the AERIS data infrastructure (https://www.aeris-data.fr/) for providing access to the IASI Level-1C data and Level-2 temperature data. The French scientists are grateful to CNES and Center National de la Recherche Scientifique (CNRS) for financial support.

## Author contributions

B.F. coordinated the study, performed the data acquisition and analyses, prepared the figures, and wrote the manuscript. L.C. designed the methodology. B.F., L.C., and P.-F.C. conceptualized the study. L.C., P.-F.C., and C.C. supervised the investigation. M.V.D. contributed to the data analyses and preparation of the figures. J.H.-L. contributed to the data processing. All authors discussed the results and revised the manuscript.

## Competing interests

The authors declare no competing interests.
