## [Peer Review File · Nature Communications]

Ethylene industrial emitters seen from spaceEditorial Note: Parts of this Peer Review File have been redacted as indicated to remove third-party material where no permission to publish could be obtained.

REVIEWER COMMENTS

Reviewer #1 (Remarks to the Author):

The manuscript by Franco et al. titled “Ethylene industrial emitters seen from space”, based on the scientific topic, definitely meets the scope of Nature Communications. Overall, the manuscript presents a novel retrieval of a trace gas and emissions, normally only detected in regions of suitable abundance (e.g. fire plumes), for industrial sources. They then go a step further to compare with bottom-up inventories. Subject to my moderate comments below, this manuscript should be suitable for publication in Nature Communications.

Moderate Comments:

- 1) The emissions are derived from long-term high resolution maps of ethylene, but would it be possible to generate e.g. annual or multi-year maps to derive emissions, so some temporal assessment of the top-down emissions can be determined?
- 2) The estimation of the ethylene is relatively simple. Would it be possible to derive a more robust estimation of the lifetime? I appreciate that a sensitivity lifetime has been tested, but I feel more needs to be done to robustly quantify this. Could surface observations or a model be used to determine this? The lifetime will also change spatially and temporally, so I believe this needs to be accounts for or some further discussion on the errors this might impose.
- 3) The discussion “supersampling” needs to be improved as I found this hard to follow.
- 4) The actual approach to estimate the mass of the source ethylene downwind is relatively crude. Instead of assuming a region around the source, could you use a simple plume model (e.g. steady state) to estimate the plume shape and then total up the mass of grid cells within it? This would help address your comment on line 567 that “we disregarded possible C₂H₄ transport out of the box” (i.e. the plume model would infer where the ethylene as gone).
- 5) Finally, the comparison with EDGAR emissions data set could be improved. While the 2010-2012 emissions might be the most up to date for that inventory, surely other inventories are more up to date? That is why I think trying to estimate top-down temporal variability in the derived emissions would improve the manuscript substantially. Also, the classification of some sources has me concerned. In the UK, for instance, the four sources are labelled as Middlesbrough, Gainsborough, Hull and Ellesmere Port with corresponding source types of petrochemical-metallurgy, coal-related industry, petrochemical and petrochemical. Firstly, for Hull, there are several large power stations upwind of it (e.g. Drax, Eggborough and Ferrybridge – least in the analysis period at least). Could these not be influencing this signal? Drax is one of the biggest emitters of air quality and GHGs in Europe, let alone the UK. For Gainsborough, this is likely from the West Burton power stations. If you can detect emissions from these power stations, then I suspect you can from Drax etc. How are the source names and industry types assigned? I just found some the sources and locations in the UK a bit surprising and if this is based on some criteria you subjectively chose, could this have knock impacts for sources from other countries?

Minor Comments:

Line 151: Can you provide some quantification of the VOC emission uncertainties. Lines 320-321: Please provide a reference.

Line 337: Might have missed this, but has ANNI been defined? Line 345: Does not make sense. Please reword.

Line 354: Do you mean that the NN is trained to miss extreme events? What would you class as extreme events? How are these defined?

Line 383-385: You say the peak concentration is fixed to the surface, but will not most power stations and other industrial sources have emissions injected into higher altitudes (e.g. several 10s-100s of metres)?

Line 380: So to clarify, the equation (could use a number) is the initial profile for ethylene? If $z_0=0$, do you need to include it in the equation?. How can sigma be random if it is set to the ERA5 boundary layer height? Why would it be restricted to 0-6 km anyway? Would be useful to state the units of sigma in its first occurrence in the manuscript. More detail on k is required as well.

Lines 385-387: Can you provide a reference for the method used.

Line 395: How do you determine if the sensitivity to ethylene is poor? How is this linked to the ratio on the next line? These few sentences could be improved.

Figure S1: Please state the grid resolution of all plots. Figure S5: Should HRI's be HRIs?

Figure S6: Is the red boxed used to the get the background in b)?

Reviewer #2 (Remarks to the Author):

The study provides estimates of ethylene emissions from industrial sources using IASI satellite observations. The authors identified 300 hotspots worldwide, and suggest that current EDGAR emission inventory underestimate industrial C₂H₄ emissions. The study is novel at providing first space-based C₂H₄ estimates, which has never been done in previous studies, and the retrieval algorithm is well designed. I think the study could be significant contribution to the field if they can provide more ground- based evidence. I also think there are some flaws in the methodology that should be addressed before publication. Below are some of my comments:

1. The first issue is that the authors didn't evaluate satellite retrieval with ground-based measurements. The reason I'm concerned about the C₂H₄ retrieval is that the retrieved C₂H₄ columns are surprisingly high (Figure S3), which are even higher than other trace gases such NO₂ and HCHO that can be

confidently retrieved from space. Even the background C₂H₄ column could reach 20×10^{15} molecules/cm², which is not a small amount. The estimated C₂H₄ for a big fire is only about 6×10^{15} molecules/cm² in Coheur et al. (2009), which made me wonder the retrieval may be biased high. There are extensive VOC measurements in US and Europe, which I think should be used to evaluate the retrieval performance. While I don't expect satellite retrieval can well capture the temporal variation of C₂H₄ from ground, the authors should at least provide evidence that their C₂H₄ estimate is within a reasonable range.

2. The emissions estimate method is not convincing to me either. The authors use a simple flux method as $E = M/\tau$. There are several issues:

1) I'm not convinced whether using a constant chemical lifetime for C₂H₄ is right. As the authors mentioned, the chemical lifetime of C₂H₄ depends on O₃ and OH level, which should vary significantly in different regions.

2) It's not clear to me why the authors didn't consider transport. For species like C₂H₄ that are relatively longer-lived, the loss against dispersion should be much larger than the loss due to chemistry. If we assume 10m/s wind speed, for the length scale of 20 km, the lifetime of C₂H₄ against dispersion is 2.7 hours, which is much shorter than the chemical lifetime of 12 hours. It's confusing that the authors rotate the observations downwind, but never accounted for wind speed in the estimates of emissions.

3) The assumption of 12 hour lifetime is not correct to me looking at Figure S3. It seems the length scale of C₂H₄ plume is around 20 km. If the lifetime is 12 hours, shouldn't it transport much further? Why is it just concentrated near source?

4) I think a better way to compare satellite-based C₂H₄ with EDGAR emission inventory is to run chemical transport models to compare model estimated C₂H₄ columns vs. satellite retrieved columns, which can avoid the issues with the emission estimates methods.

3. While it's interesting to provide high resolution estimates of C₂H₄, it's not clear to me what are the values of high-resolution data. The authors actually didn't take advantage of the spatial patterns of C₂H₄ in the emission estimates, and they simply take the spatial average of C₂H₄ of a big box. If so, why do they even need high spatial resolution?

4. The results section are overall qualitative. Figure 2 only shows the distribution of the hotspots, but what the concentration levels? How do they vary spatially? I actually found Figure S2 more informative. Especially since the next section only focused on 53 out of 300 hotspots, there is almost no quantitative information about the other hotspots.

Minor comments:

Figure 1: Better label the location of the sources on the Google Earth images.

Figure 3: Table S2 suggests large variation of IASI flux with lifetime, but the uncertainty of the C₂H₄ flux is not shown here. I'd suggest include an uncertainty estimates here.

Figure S3: It's not clear which is the downwind direction.

References:

Coheur, P.-F., Clarisse, L., Turquety, S., Hurtmans, D., and Clerbaux, C.: IASI measurements of reactive trace species in biomass burning plumes, *Atmos. Chem. Phys.*, 9, 5655–5667, <https://doi.org/10.5194/acp-9-5655-2009>, 2009.

Reviewer #3 (Remarks to the Author):

This article presents ethylene observations made from the IASI instruments on the Metop satellites. Using IASI and advances in retrieval algorithms and long-term averaging, the authors build a high- resolution global dataset of ethylene at 0.01x0.1 degree resolution (~1km) averaged for 2008-2020. This is done using a wind-adjust super-resolution oversampling, neural networks, and a recently developed “whitening” spectral technique (all detailed thoroughly in the text). I commend the authors for their analysis. They are really pushing the data to its limits.

Using this dataset, they are able to detect 336 ethylene emission global hotspots. To my knowledge, this is the first time industrial emissions of ethylene have been detailed from space (there has been previous detection in biomass burning plumes) and the number they have detected is impressive. Ethylene flux is quantified for 57 large hotspots. Of significance, the authors find the widely-used emissions database EDGAR underestimates ethylene often by 1-2 orders of magnitude and completely misses 35 of 53 industrial sources.

The paper shows a large range of fluxes assuming different lifetimes (see Table 2 in the Supplement) and points to the need to quantify lifetimes of ethylene for various conditions, and the need to improve speciation knowledge of VOC emissions in inventories. As a result, there is still a lot of work to do in field measurements and modeling to make this data really useful for top-down (satellite-based) emissions inventories. Overall the mapping of ethylene hotspots is an interesting development in and of itself, but the results also point to some major uncertainties in our current inventories and understanding of global VOC emissions.

The paper presents exciting results which will be of significant interest to several communities in atmospheric science, including those working on trace gas retrievals from space and atmospheric

modellers. It will also be of significant interest to regulatory agencies who depend on accurate emissions inventories, and to the general public and policy makers who are interested in pinpointing sources of industrial pollutants. I would recommend this work be published in any journal and I believe it of high quality and broad enough interest for Nature Communications.

Specific Comments:

Line 36: Also early detection by Alvarado et al. (2011) using TES:
<https://doi.org/10.3390/atmos2040633>

Line 72: Can you be specific here about online sources? Is this all done by visual interpretation and then looking up potential sources? It's a bit vague but I'm guessing this is how it's done?

Line 321: It's not clear to me until much later in text... Are you using merged data from all these three IASI instruments? Are they consistent with each other during overlapping periods? Spatial resolution of IASI is mentioned later but I think it should also be introduced here with other instrument information.

Line 364: Probably should refer to a figure here to give evidence of this statement.

Line 384: Is the peak concentration fixed at the surface everywhere, or does it vary away from a priori sources? I imagine it's the first as this is in the training set development but can you please clarify here.

Line 445: Presumably there is quite a lot of temporal variability in 13 years that you don't detect. How does this affect the estimates and super resolution calculation? For instance, emitters have been changing rapidly in a place like China. Could there be large uncertainties from plants coming on and offline? How is this dealt with the EDGAR comparisons?

Line 470: "all the uncertainties associated with the retrieved column" : Perhaps these uncertainties have been previously discussed, but feel like I need to be reminded here of what they are. Also here or elsewhere, can you say a few more words about quantitative uncertainties? What are the errors introduced by the vertical sensitivity of IASI?

Line 575: What is the background ethylene typically removed? How different is this from modelled background ethylene?

Supp. Fig 1: Is wind-rotated oversampling and super-sampling showing all ethylene to the east because this direction is considered “downwind”? Might be useful to note this somewhere.

Supp. Fig 2: I find the color scheme superimposed on the global visible imagery difficult to read (for example, shades of green on green, and is that white on a dark blue ocean? – I can’t tell). I find it fine to look at in the very high resolution imagery. However, there are less contrasts in the background of the global imagery which is maybe why it is difficult to see what is ethylene and what is the surface.

Technical Comments:

Line 14: Awkward sentence, suggest change to “Here, we track from space over 300 worldwide hotspots of ethylene, the most abundant industrially produced organic compound.”

Line 31: Similar awkward sentence, “which contributes” refers to ethylene and not “sources” so needs to come earlier after the sentence subject, or change to “and contributes”.

Line 35: Same sentence structure issue, “dominated by natural sources” does not refer to troposphere but to “concentration”

Line 52: “We took benefit of the” doesn’t make sense and sentence is a bit awkward.

Line 320: “embarked” not really used in this context. Suggest “flying on”

Line 345: Change to “among which are included the VOCs”

Line 352: “built in such a way”

Line 425: Change “took benefit of” to “took advantage of”

Line 568: “In such a way”

Ethylene industrial emitters seen from space

Franco et al. - NCOMMS-22-11986

Response to the reviewers

Reviewer #1 (Remarks to the Author):

The manuscript by Franco et al. titled “Ethylene industrial emitters seen from space”, based on the scientific topic, definitely meets the scope of Nature Communications. Overall, the manuscript presents a novel retrieval of a trace gas and emissions, normally only detected in regions of suitable abundance (e.g. fire plumes), for industrial sources. They then go a step further to compare with bottom-up inventories. Subject to my moderate comments below, this manuscript should be suitable for publication in Nature Communications.

We would like to thank the referee for the detailed review and the comments that contributed to improving the manuscript. In particular, the referee motivated us to attempt a temporal assessment of the C₂H₄ hotspots and of their top-down emission fluxes. This assessment led to interesting results that are now included in the manuscript. Please find in blue here below our response to the comments and the changes made to the manuscript.

Moderate Comments:

1) The emissions are derived from long-term high resolution maps of ethylene, but would it be possible to generate e.g. annual or multi-year maps to derive emissions, so some temporal assessment of the top-down emissions can be determined?

We thank the referee for this suggestion. In the revision, we have now investigated whether the IASI data can be used to reveal temporal changes of C₂H₄ emissions. For this, we applied the wind-rotated supersampling to the IASI measurements over the 2008-2011, 2012-2014, 2015-2017 and 2018-2020 time periods. Analysis of the multi-year maps reveals that the representation of many C₂H₄ point-sources, especially the weak hotspots, is significantly more difficult because of fewer satellite data and, as a result, of lower signal-to-noise ratios.

Nonetheless, for a series of major hotspots, the representation of the point-sources and transport plume remains good enough in the 3(4)-year high-resolution maps to allow us to assess their top-down C₂H₄ emissions (using the same approach as detailed in the Methods). Relatively stable emission fluxes were derived for most of the major hotspots during the 4 multi-year periods. However, interestingly, some point-sources do exhibit a significant increase over time of the C₂H₄ HRI hotspot and associated emission fluxes. Two examples are displayed in the figures here below: the first one shows the point source of Hainan (Inner Mongolia, China) concentrating numerous coal-related activities, and the second one, the large petrochemical hub of Yanbu (Saudi Arabia). In both cases, the increase of the observed emission fluxes is corroborated by a densification of industrial complexes seen in satellite imagery taken within each period. We added the figure associated to the first example to the main manuscript (and the second one to the Supplementary Information) and added a discussion of the temporal aspects as reported here below.

[Redacted]

[Redacted]

“To investigate whether IASI can reveal temporal changes in the anthropogenic emissions of atmospheric C₂H₄, we applied the wind-adjusted super-resolution technique to the satellite data over four different time periods: 2008-2011, 2012-2014, 2015-2017 and 2018-2020. Despite the decrease in the signal-to-noise ratio, the representation of the point-sources and transport plumes is good enough to allow studying the time evolution of the largest hotspots. For most, no significant trends were observed, but for others, the hyperfine resolution maps reveal a clear progressive enhancement of the downwind C₂H₄ HRI average with time, as illustrated in Fig. 5 for a coal-related hotspot in Inner Mongolia, Central China. Another example over a large petrochemical hub in Saudi Arabia is presented in Supplementary Fig. 10. For these two examples, the top-down C₂H₄ emissions grow respectively from 7.2×10^{-2} and 8.8×10^{-2} kg s⁻¹ for the 2008-2011 period, to 2.6×10^{-1} and 2.9×10^{-1} kg s⁻¹ over 2018-2020. Satellite visible imagery supports this finding, as new industrial complexes are seen to appear over the different periods (Fig. 5, Supplementary Fig. 10). Similar examples are found in the central and western regions of China, which have recently undergone programs of industrial development that rely heavily on the exploitation of local coal resources as raw materials and for energy production^{25, 26}. Currently, the temporal assessment is limited to the most prominent point-sources and to multiyear time blocks. These examples, however, represent the first demonstration of how infrared sounders can be used to monitor industrial emissions of VOCs, a capability that is surely going to improve with the next-generation infrared sounders that will offer better spectral resolution and lower noise in case of IASI- NG³³, and increased temporal sampling in case of the geostationary infrared sounder onboard the Meteosat Third Generation (MTG-IRS) satellite (<https://www.eumetsat.int/meteosat-third-generation>).”

2) The estimation of the ethylene is relatively simple. Would it be possible to derive a more robust estimation of the lifetime? I appreciate that a sensitivity lifetime has been tested, but I feel more needs to be done to robustly quantify this. Could surface observations or a model be used to determine this? The lifetime will also change spatially and temporally, so I believe this needs to be accounts for or some further discussion on the errors this might impose.

We agree with the referee that using a constant chemical lifetime of C₂H₄ is a simplification. However, we believe that it is justified considering the current lack of global and consistent constraints. Indeed, global models are of little usefulness for such endeavour because of their too coarse horizontal resolution relative to the spatial extent of the C₂H₄ point-sources, i.e., a typical hotspot of 20-30 km extent is largely “diluted” in a model grid cell at a 1° × 1° resolution (corresponding roughly to the highest model resolution). For instance, global models would be unable to represent the high concentrations in atmospheric oxidants (mainly OH and O₃), significantly reducing the atmospheric lifetime of C₂H₄, as measured in the vicinity of the anthropogenic point-sources. Moreover, estimates of the C₂H₄ lifetime based on in situ data, mostly aircraft measurements in industrial plumes, are scarce in both time and space. For example, Cho et al. (2021), Washenfelder et al. (2010) and Wert et al. (2003) derived C₂H₄ lifetimes of a few hours during such aircraft campaigns over specific regions of Texas and South Korea. Considering the scarcity of these measurements, it is highly challenging to extrapolate such short lifetimes to all the point-sources worldwide for which we derived emission fluxes from IASI. Therefore, we assumed in our work a constant, more conservative lifetime (12 h), while using a shorter lifetime (2 h) as an upper-bound estimate of the C₂H₄ emission fluxes that could be derived from IASI (it is worth noting that in all cases the main conclusions of the study remain valid). While this approach is crude, it allows putting robust upper and lower bounds on the derived emissions. This is also the reason why such an approach has been heavily favoured in previous state-of-the-art studies to derive emission fluxes worldwide from localized anthropogenic point-sources of SO₂ (Carn et al., 2007; Fioletov et al., 2011, 2013; Li et al., 2017; McLinden et al., 2016; Wang et al.,

2015), NO₂ (Ghude et al., 2013), and NH₃ (Van Damme et al., 2018).

A major step forward would consist in estimating the atmospheric C₂H₄ lifetime for each point-source, based directly on the satellite measurements themselves (e.g., based on the extent of the pollution plume, as done for SO₂ by Fioletov et al., 2015, 2016, and for NO₂ by Beirle et al., 2011). However, this approach is currently not workable for C₂H₄, as it requires satellite measurements with very low noise, over both source and remote regions. With the future launch of IASI-NG, which will have much higher instrumental performance compared to its predecessor, such an approach will be more likely attainable.

We now acknowledge in the Methods that a constant lifetime is a simplification, and that future work and satellite measurements will be needed to address this:

“Assuming a constant lifetime is a simplification in the calculation of the top-down fluxes. To account for the spatial and temporal variability of the atmospheric C₂H₄ lifetime, a step forward would consist in estimating this lifetime for each point-source, based directly on the satellite measurements themselves, as done for SO₂^{46,52} and for NO₂¹. However, this approach is currently not workable for C₂H₄, as it requires satellite measurements with very low noise, over both source and remote regions. With the future launch of IASI-NG, which will have much higher instrumental performance compared to its predecessor³³, such an approach will be more likely attainable.”

3) The discussion “supersampling” needs to be improved as I found this hard to follow.

We now provide a more detailed explanation on the supersampling technique in the Methods, and we hope it is now easier for the reader to follow. It is difficult to expand further the summary of the supersampling without providing the full detailed explanation as provided in Clarisse et al. (2019). However, the reference is available to those readers who want to understand the method on a deeper level. The new discussion reads:

“Super resolution. Described by Ref. 23, the supersampling (or super-resolved oversampling) is derived from super-resolution techniques as it aims at reconstructing high-resolution images from numerous low-resolution representations of the same scene. It consists in repeating the oversampling procedure and in correcting, at each iteration, the resulting oversampled average according to the differences with the ground truth (the satellite observations). Specifically, the solution of the supersampling for the first iteration corresponds to the oversampling. Then, we calculate the differences between the single-pixel IASI measurements and the same individual observations simulated by assuming that this first-iteration supersampling is the ground truth. For the second iteration, the solution consists in adding the oversampled average of these measurement differences to the supersampling solution from the previous step. This procedure is repeated until a supersampled average that is optimally consistent with the ground truth is found. A condition for the application of this technique is to assume that the underlying satellite data distribution is relatively constant in time. As demonstrated by Ref. 23, redistributing the daily IASI measurements according to the wind fields, as done with the wind rotation presented above, allows removing most of the variability over a gas point-source and guarantees the required homogeneity. Supplementary Figs 1-2d show examples of downwind average of C₂H₄ HRI obtained by applying the supersampling to the daily wind-rotated IASI maps around a point-source. While the oversampling typically smooths out the satellite data, the supersampling reproduces more realistically the strength of the hotspots and resolves much finer spatial features (of about 3-4 km with IASI). This allows the discovery of hotspots that would be difficult to find with the oversampling techniques, as illustrated in Supplementary Fig. 2. For instance, it allowed doubling the number of NH₃ point-sources identified with IASI²³ compared to those detected with oversampling³. It is worth noting that the super-resolution preserves the gas mass around the point-source with respect to the original IASI data, as

shown by Ref. 23.”

4) The actual approach to estimate the mass of the source ethylene downwind is relatively crude. Instead of assuming a region around the source, could you use a simple plume model (e.g. steady state) to estimate the plume shape and then total up the mass of grid cells within it? This would help address your comment on line 567 that “we disregarded possible C₂H₄ transport out of the box” (i.e. the plume model would infer where the ethylene as gone).

We agree that the proposed approach would work well. However, we do not clearly see its advantage compared to the method implemented in the study. Both approaches aim at estimating the mass within the plume, with a possible background correction. In our approach, the sum is made over the entire box, with a background contribution subtracted over the entire box. Mathematically, this is equivalent by summing up only over the alleged plume corrected for by a background offset. In both approaches, correctly assessing the background is key.

Note that the employed method does consider fully the contribution of transported C₂H₄ as long as the box size is sufficiently large, and within the constraints of the satellite retrieval detection limit. This is already stressed in the text: “*For each hotspot, the size of the box was adjusted in such a way to encompass most of the transport plume and minimize the amount of ethylene transported out of the area.*” Note also that our approach assumes steady state (the condition for which $E=M/\tau$ is valid).

5) Finally, the comparison with EDGAR emissions data set could be improved. While the 2010-2012 emissions might be the most up to date for that inventory, surely other inventories are more up to date? That is why I think trying to estimate top-down temporal variability in the derived emissions would improve the manuscript substantially. Also, the classification of some sources has me concerned. In the UK, for instance, the four sources are labelled as Middlesbrough, Gainsborough, Hull and Ellesmere Port with corresponding source types of petrochemical-metallurgy, coal-related industry, petrochemical and petrochemical. Firstly, for Hull, there are several large power stations upwind of it (e.g. Drax, Eggborough and Ferrybridge – least in the analysis period at least). Could these not be influencing this signal? Drax is one of the biggest emitters of air quality and GHGs in Europe, let alone the UK. For Gainsborough, this is likely from the West Burton power stations. If you can detect emissions from these power stations, then I suspect you can from Drax etc. How are the source names and industry types assigned? I just found some the sources and locations in the UK a bit surprising and if this is based on some criteria you subjectively chose, could this have knock impacts for sources from other countries?

To the best of our knowledge, EDGAR v4.3.2 is one of the few existing anthropogenic inventories providing emission fluxes specific to C₂H₄, at a fairly high spatial resolution (0.1°×0.1°). Furthermore, it is by far the most comprehensive in terms of emission sectors, emitted species, speciation profiles and time resolution, and the most widely used anthropogenic inventory in the atmospheric science community. For example, other state-of-the-art global inventories such as CEDS (Community Emissions Data System; McDuffie et al., 2020) and ECLIPSE (Evaluating the Climate and Air Quality Impacts of Short-Lived Pollutants; Stohl et al., 2015) have a lower spatial resolution (0.5°×0.5°) and do not provide fluxes specific to C₂H₄ (all the NMVOCs are treated as a single compound). A big advantage of EDGAR over regional inventories is that it provides VOC emission fluxes calculated in a fully consistent way throughout the globe. This consistency worldwide is a real strength as it allows a global assessment of the C₂H₄ industrial emissions with the IASI-based fluxes. For these reasons, we believe that EDGAR is the best choice in the framework of our study. Of course, comparing the IASI-based emission fluxes with regional inventories would be interesting and could be part of future works dedicated to the industrial emissions in specific regions.

Regarding the temporal variability of the top-down emissions, we agree that it improves the manuscript. On that point, we refer to our answer to the first comment. Unfortunately, the speciated VOC emissions from EDGAR being available until 2012, no conjoint temporal evolution of the EDGAR and IASI fluxes can be studied.

We understand the referee's concern with respect to the identification of the point-sources and to the assignment of their names and industry types. Here below, we clarify the procedure we followed, and we provide more information on the UK hotspots:

- As shown by Suppl. Figs 1-2, the wind-rotated super-resolution contributes to enhancing the contrast between the point-source and the adjacent background. Although a C₂H₄ hotspot has some spatial extent (typically 20-30 km, as illustrated in Suppl. Figs 1-3), another advantage of the supersampling is to reproduce more realistically the strength of the hotspot as opposed to the smoothed distribution obtained with the oversampling (see Suppl. Figs 1-2). As a result, it is much easier to identify the hotspot centre and to identify potential emitter(s) underneath with satellite imagery. Moreover, the location of the C₂H₄ point-sources is calculated by an algorithm providing the exact coordinates of the highest local HRI value in the hyperfine distribution for each detected hotspot. On a subset of known NH₃ emitters, it was shown by Clarisse et al. (2019) that the source could be located within a median distance of 1.5 km. Let us also recall here that the wind-rotated supersampling preserves the distance to the point-source (Clarisse et al., 2019). Therefore, the C₂H₄ emitter must be found within a few km from the hotspot centre, whereas potential emitters located further can be excluded (as it is the case for the Brax, Eggborough and Ferrybridge power plants; see below). Inside that small area, with the help of satellite imagery and online information we list all the anthropogenic activities that are known to emit C₂H₄, which overall fall into four categories (petrochemistry, metallurgy, coal-related activities and extended, dense urban areas). When no or insufficient information is found on the potential emitter(s), we use the "undetermined" category to classify the point-source.
- A hotspot is usually named according to the closest major settlement. For example, in the case of Middlesbrough, the hotspot is located to the east of the city where are located a steel factory of British Steel (the Teesside Steelworks) and a petrochemical cluster hosting, e.g., Sabic UK which runs a least one ethylene cracker in that site. Because of the close vicinity between the steel factory and the petrochemical activities, it is impossible for us to distinguish the main contributor to the hotspot. Therefore, both categories are attributed to Middlesbrough. Here, it is important to note that the size of Middlesbrough city is way too small to be considered as a major C₂H₄ emitter in comparison to the urban point-sources we detect around the world and that are associated with megacities (e.g., Mexico City, Tehran, Jakarta). Regarding Ellesmere Port, the hotspot is located just north of this location and encompasses several (petro)chemical hubs located on both banks of the River Mersey.
- As for the assignment of Hull and Gainsborough hotspots: To the southeast of Hull hotspot (at <4 km) is the Saltend Chemicals Park, a petrochemical cluster hosting companies such as Ineos and British Petroleum, and most likely the principal contributor to the C₂H₄ hotspot. There are indeed several coal-fired power plants to the west of Hull, but those are located at >40 km (Brax), >50 km (Eggborough) and >60 km (Ferrybridge) from the detected hotspot, i.e., too far to contribute to the C₂H₄ enhancement over Hull. In addition, we do not observe any clear hotspot close to these power plants. Regarding Gainsborough hotspot, we indeed attribute it to the West Burton power stations located just underneath the hotspot centre. We agree, it is intriguing that IASI

detects this point-source but does not show significant C₂H₄ enhancements over Drax, Eggborough and Ferrybridge. It is sometimes unclear why some

point-sources can be detected, whereas others of the same type cannot. For the undetected hotspots, a reasonable explanation (other than more controlled emissions) is that local and frequent meteorological factors prevent the pollution plume to be brought at higher altitudes where the detection by IASI is easier. As a result, the C₂H₄ enhancement does not show enough contrast with the background, and we cannot claim with confidence the existence of a point-source there.

Minor Comments:

Line 151: Can you provide some quantification of the VOC emission uncertainties.

Unfortunately, it is impossible for us to provide a range of uncertainty on the VOC emissions from EDGAR, considering the several steps and assumptions that are made to obtain spatially resolved, sector-specific fluxes of C₂H₄. As explained in Huang et al. (2017), the C₂H₄ fluxes are typically speciated from total NMVOC emissions available in existing databases (e.g., national inventories). Just the uncertainties on such databases are very challenging to gauge. Another large source of uncertainties is the calculation and application of the sector-specific speciation profiles to disaggregate the C₂H₄ fluxes from the total NMVOC emissions (Huang et al., 2017). It is almost impossible to keep track of the error propagation through all these complex processes. Other sources of uncertainties include, for instance, spatial gridding of the emission fluxes.

As we are unable to quantify the VOC emission uncertainties from EDGAR, we acknowledge that our initial sentence in the manuscript is not appropriate. To better reflect what is explained above, we have reworded it as follows:

“Consequently, numerous uncertainties add up in the bottom-up estimates of VOC emissions.”

Lines 320-321: Please provide a reference.

We have added a reference to Clerbaux et al. (2009), who describe IASI and its application to remote sensing of the atmospheric composition.

Line 337: Might have missed this, but has ANNI been defined?

Indeed, ANNI (Artificial Neural Network for IASI) has been defined at the beginning of the same paragraph.

Line 345: Does not make sense. Please reword. We have reworded the sentence

as follows:

“This detection method is proven to be very sensitive and well suited for the detection of the VOCs³⁵⁻³⁸, which are characterized by weak and broadband absorptions in the thermal infrared.”

Line 354: Do you mean that the NN is trained to miss extreme events? What would you class as extreme events? How are these defined?

Our initial statement can indeed be misleading. We meant here that the NN, during its training phase, must be presented with enough examples with, e.g., very high gas abundance or very large thermal contrast, to ensure it has been trained to provide realistic and robust results even when “extreme” atmospheric conditions it has not seen before, are met (e.g., in fire plume). It is usually referred to as the generalizability of the NN. To clear up misunderstanding, we have amended the sentence as follows:

“The trained NN is tested carefully to ensure that it generalizes well, i.e., that it is able to provide realistic results for all abundances of the target species and states of the atmosphere encountered in real observations.”

Line 383-385: You say the peak concentration is fixed to the surface, but will not most power stations and other industrial sources have emissions injected into higher altitudes (e.g. several 10s-100s of metres)?

We fully agree with the referee, and in the manuscript, this is one of the reasons we invoked to explain why, in general, industrial emissions of C₂H₄ seem to be easier to detect with IASI than urban emissions. However, the injection height and subsequent transport is highly dependent on local meteorological conditions (e.g., vertical temperature profile, wind speeds...) and varies in time. It is therefore challenging to attribute for a given site – via, e.g., a model of plume dispersion - an injection height that is representative for the entire IASI time series. Moreover, such estimate of the injection height should be done in a fully consistent way for all the C₂H₄ point-sources that are studied here. This makes the task even more complex to achieve in the framework of this study. Finally, from the two C₂H₄ vertical profiles obtained with equation R1 in the Methods, assuming a C₂H₄ vmr of 5 ppb at the altitude of the peak and the 1976 US standard atmosphere, we calculate corresponding C₂H₄ total columns of 2.13×10^{16} and 2.64×10^{16} molecules cm⁻² when this peak is at 0 km and 0.5 km altitude, respectively. This represents roughly a 20% difference, which is typically well below the 1-2 order(s) of magnitude of difference that we observe between the IASI-derived and the EDGAR emission fluxes over the industrial point-sources. Therefore, although it would certainly be valuable to estimate the injection height for future works, it does not change the main conclusions of the present study.

We have added this information to the Methods, in the section “C₂H₄ retrieval and IASI dataset”:

“To evaluate the error on the IASI vertical profile, we used R1 to build two C₂H₄ vmr profiles assuming a peak concentration of 5 ppb (vmr(C₂H₄) at z₀) at, respectively, surface and 0.5 km altitude. Using the 1976 US standard atmosphere, we calculated C₂H₄ total columns of 2.13×10^{16} and 2.64×10^{16} molecules cm⁻² from these two profiles. This 20% difference is well below the typical 1-2 order(s) of magnitude of discrepancy that we observe between the top-down and EDGAR emissions over industrial point-sources.”

Line 380: So to clarify, the equation (could use a number) is the initial profile for ethylene? If z₀=0, do you need to include it in the equation? How can sigma be random if it is set to the ERA5 boundary layer height? Why would it be restricted to 0-6 km anyway? Would be useful to state the units of sigma in its first occurrence in the manuscript. More detail on k is required as well.

The parametrization of the C₂H₄ vertical profile is directly adapted from the procedure of NH₃ retrieval from IASI (Whitburn et al., 2016; Van Damme et al., 2021). Following this procedure, the NN set up specifically for C₂H₄, has been trained for peak heights of the profile located not only at the surface (z₀ = 0 km), but also in altitude. This allows us to retrieve C₂H₄, e.g., in a concentrated fire plume, using the same NN. Similarly, sigma (in km) – which represents the standard deviation of the Gaussian - has been varied during the training process to ensure the NN can account for different thicknesses of the PBL (when z₀ = 0 km) or different thicknesses of a fire plume (when z₀ is in altitude). The goal is to make the C₂H₄ NN as flexible and comprehensive as possible. Regarding the scaling factor k (expressed in ppb), it allows us to adjust the peak of the parameterized profile (i.e., of the Gaussian) to the desired C₂H₄ vmr. During the training process, k is varied to train the NN from very low to very large abundance

of the target species. In the Methods, we have cleared up the explanation on the equation parameters:

“with σ_0 the peak height of the vertical profile (in km) as a measure of the altitude of the bulk of C_2H_4 ,

σ the standard deviation of the Gaussian function as a measure of the thickness of the C_2H_4 layer (in km; assigned as explained below), and α (in ppb) a scaling factor of the profile that controls $vmr_{(C_2H_4)}$ at σ_0 and hence the C_2H_4 abundance in the forward simulations. In this work, the peak concentration was always fixed at the surface to be representative for observations over hotspots, as the bulk of C_2H_4 is assumed to be close to the surface.

[...]

The value of α was set to the ERA5 boundary layer height.”

Lines 385-387: Can you provide a reference for the method used.

We have added a reference to Van Damme et al. (2021) for NH_3 , and to Franco et al. (2018, 2019, 2020) for the VOCs.

“The NN trained for ethylene is made of two computational layers of 12 nodes; a satisfactory performance similar to that of NH_3 ³⁵ and other VOCs³⁶⁻³⁹ was indeed reached with this network architecture, while keeping the NN relatively small to prevent overfitting.”

Line 395: How do you determine if the sensitivity to ethylene is poor? How is this linked to the ratio on the next line? These few sentences could be improved.

By definition (see, e.g., Walker et al., 2011; Whitburn et al., 2016; Franco et al., 2020), the HRI quantifies the magnitude of the absorption of a target gas within a spectral range of an individual observation. It is a (dimensionless) metric of the signal intensity captured by the satellite and directly related to the gas abundance. In particular, for a given state of the atmosphere and surface, a higher (lower) HRI corresponds to a higher (lower) gas column. As the HRI is the link between the spectrum and the gas abundance, the HRI value also varies according to the state of the atmosphere and surface (mainly the thermal contrast). Specifically, for a constant column, the HRI has a higher (lower) value for a larger (weaker) thermal contrast. Or, for a constant HRI, a weaker (larger) thermal contrast translates in a higher (lower) column. As the data post-filtering applied here is based on the column- to-HRI ratio, it is kept independent (in first order) from the gas abundance. This means that in practice, the thermal contrast (i.e., the surface and atmospheric conditions of the observation scene) is the main driver of the post-filtering and not the column. As shown by Whitburn et al. (2016) and Franco et al. (2020), it aims at removing unphysical retrievals as it excludes measurements for which a large HRI is observed corresponding to an almost zero thermal contrast. Such conditions are met when there are large errors associated with the parameters describing the state of the atmosphere and surface (e.g., temperature profile, surface emissivity artefacts), hence when bad observational conditions deplete the ability of IASI to detect the target species. In such a case, we can say that the IASI sensitivity to C_2H_4 is poor.

We have clarified the explanation on the post-filtering as follows:

“Consistent with the other products of the ANNI framework, a pre-filter prevents the retrieval on cloudy scenes and a post-filter discards the C_2H_4 columns affected by too large uncertainties or by poor observational conditions to retrieve ethylene. In practice, the retrieved column is rejected when the

ratio $|\text{column}_{(C_2H_4)} / \text{HRI}_{(C_2H_4)}|$ is higher than 5×10^{16} molecules cm^{-2} or when the spectral baseline temperature is lower than 265 K. It was shown that this type of post-filter, based on the column-to-HRI ratio, is primarily driven by the thermal contrast of the observation scene, i.e., by the surface and atmospheric conditions, and not directly by the gas abundance^{34, 38}. Typically, it allows removing unphysical retrievals when poor observational conditions limit

the ability of IASI to detect the target species.”

Figure S1: Please state the grid resolution of all plots.

The grid resolution is provided in the figure caption: $0.15^\circ \times 0.15^\circ$ for panel **a**, and $0.01^\circ \times 0.01^\circ$ for panels **b-d**. In km, at the latitude of the hotspot displayed in this figure (39.9° N), this corresponds to a grid resolution of 12.8×16.6 km (lon \times lat) for panel **a**, and of 0.9×1.1 km for panels **b-d**. We have added this information to the figure caption. We have also added the scale to all plots.

Figure S5: Should HRI's be HRIs?

Corrected.

Figure S6: Is the red boxed used to the get the background in b)?

The area used to calculate the C_2H_4 background column corresponds to the semi-transparent area in panel **a**. The pixels inside the area delimited in red are used to compute the point-source emissions from the EDGAR inventory.

Reviewer #2 (Remarks to the Author):

The study provides estimates of ethylene emissions from industrial sources using IASI satellite observations. The authors identified 300 hotspots worldwide, and suggest that current EDGAR emission inventory underestimate industrial C₂H₄ emissions. The study is novel at providing first space-based C₂H₄ estimates, which has never been done in previous studies, and the retrieval algorithm is well designed. I think the study could be significant contribution to the field if they can provide more ground-based evidence. I also think there are some flaws in the methodology that should be addressed before publication.

We would like to thank the referee for the review and the comments, which stimulated us to perform a series of additional analyses and eventually resulted in a better manuscript. In particular, the comments encouraged us to perform a first comparison of the IASI C₂H₄ measurements with independent surface and aircraft data. This comparison gives confidence in the satellite data over the C₂H₄ hotspots and is now included in the study. Please find in blue here below our response to the comments and the changes made to the manuscript.

Below are some of my comments:

1. The first issue is that the authors didn't evaluate satellite retrieval with ground-based measurements. The reason I'm concerned about the C₂H₄ retrieval is that the retrieved C₂H₄ columns are surprisingly high (Figure S3), which are even higher than other trace gases such NO₂ and HCHO that can be confidently retrieved from space. Even the background C₂H₄ column could reach 20×10^{15} molecules/cm², which is not a small amount. The estimated C₂H₄ for a big fire is only about 6×10^{15} molecules/cm² in Coheur et al. (2009), which made me wonder the retrieval may be biased high. There are extensive VOC measurements in US and Europe, which I think should be used to evaluate the retrieval performance. While I don't expect satellite retrieval can well capture the temporal variation of C₂H₄ from ground, the authors should at least provide evidence that their C₂H₄ estimate is within a reasonable range.

We agree with the referee that a comparison between the IASI C₂H₄ retrievals and independent measurements is desirable. Unfortunately, representative column-to-column comparisons are currently not within reach as only few ground-based FTIR C₂H₄ total column measurements exist (Vander Auwera et al., 2014; Toon et al., 2018), and those are currently too scarce and essentially limited to remote areas to enable a thorough evaluation of the IASI retrievals. In such background conditions, the low C₂H₄ level is indeed close to the detection threshold in the IASI spectra (which is also true for the ground-based FTIR spectra), hence we cannot rule out a potential bias in the IASI retrievals. This is the reason why we applied the super-resolution technique to the HRI (the HRI not being affected by extra uncertainties on the retrieved columns) for searching hotspots, and why we retrieved C₂H₄ abundances in the vicinity of hotspots only.

However, as we agree that it is important to show that the retrieved columns are reasonable, we went through the literature and found for a selection of hotspots nearby surface in situ or low-altitude aircraft measurements of C₂H₄ concentrations. We then converted these (near-) surface data in C₂H₄ total columns by assuming the same vertical distribution of C₂H₄ as the Gaussian function implemented by the ANNI v3 neural network to retrieve C₂H₄ columns from IASI (see Methods). To keep this analysis straightforward and synthetic, we assumed in all cases the 1976 US standard atmosphere and a planetary boundary layer height at 1 km altitude. For a suite of IASI hotspots, we report in the table here below the range of IASI columns retrieved over these hotspots, the range of C₂H₄ volume mixing ratios found in the literature and the corresponding total columns we calculated.

The comparison shows that the IASI C₂H₄ columns fit well within the ranges of abundance derived from the independent measurements and gives confidence in the satellite retrievals. We have added this table to the Supplementary Information (Tab. 3) and introduced it in the Methods (section “C₂H₄ retrieval and IASI dataset”).

Hotspot	In situ VMR (ppb)	In situ columns (molecules cm ⁻²)	IASI columns (molecules cm ⁻²)	Reference(s)
Dagang (Tianjin, China)	2.6-33.1	0.8-9.8 × 10 ¹⁶	1.5-2.5 × 10 ¹⁶	Wei et al. (2018)
Daesan (South Korea)	2.6-26.5	0.8-7.8 × 10 ¹⁶	1.5-2.5 × 10 ¹⁶	Cho et al. (2021) Simpson et al. (2020)
Dushanzi (Xinjiang, China)	1.0-2.4	0.3-0.7 × 10 ¹⁶	0.8-1.0 × 10 ¹⁶	Zhang et al. (2019)
Fangshan (Beijing, China)	5.6	1.7 × 10 ¹⁶	2.0-3.0 × 10 ¹⁶	Wei et al. (2015)
Houston (Texas, USA)	1.4-10.0	0.4-2.9 × 10 ¹⁶	1.0-2.0 × 10 ¹⁶	De Gouw et al. (2009) Johansson et al. (2014) Ryerson et al. (2003) Wert et al. (2003)
Kaohsiung (Taiwan)	6.7-19.9	2.0-5.9 × 10 ¹⁶	2.0-3.0 × 10 ¹⁶	Chang et al. (2005)
Lanzhou (Gansu, China)	2.9-10.8	0.9-3.2 × 10 ¹⁶	2.0-3.0 × 10 ¹⁶	Jia et al. (2016) Wu et al. (2019)
Mexico City (Mexico)	10.0-37.0	3.0-17.7 × 10 ¹⁶	1.0-2.0 × 10 ¹⁶	Altuzar et al. (2005) Velasco et al. (2007)
Nanjing (Jiangsu, China)	5.7	1.7 × 10 ¹⁶	2.0-3.0 × 10 ¹⁶	An et al. (2014)
Saõ Paulo (Brazil)	4.0-5.6	1.2-1.6 × 10 ¹⁶	0.5-2.0 × 10 ¹⁶	Alvim et al. (2018) Dominutti et al. (2016)
Ulsan (South Korea)	7.8-29.1	2.3-8.6 × 10 ¹⁶	1.5-2.5 × 10 ¹⁶	Na et al. (2001)
Wuhan (Hubei, China)	7.8-15.3	2.3-4.5 × 10 ¹⁶	1.5-3.0 × 10 ¹⁶	Shen et al. (2018) Zheng et al. (2020)
Yokohama (Tokyo Bay, Japan)	3.0-11.2	0.9-3.3 × 10 ¹⁶	1.5-2.5 × 10 ¹⁶	Tiwari et al. (2010)

It is interesting to note that elevated C₂H₄ total columns as we retrieve above hotspots (> 1×10¹⁶ molecules cm⁻²) are not rare and have already been retrieved from both IASI and ground-based FTIR measurements in fire plumes. In Coheur et al. (2009), C₂H₄ columns of 6×10¹⁵ molecules cm⁻² were indeed retrieved from IASI observations during the 2007 Greek fires in Peloponnese, but in an aging plume located over the Mediterranean Sea, at a distance from the fire source (Fig. 6; Coheur et al., 2009). Considering the short C₂H₄ lifetime, a significant part of the emitted C₂H₄ is expected to have already been removed from the atmosphere. Indeed, during the same fire event, C₂H₄ total columns in the range of 1×10¹⁷ molecules cm⁻² were retrieved closer to the source (Fig. 8; Coheur et al., 2009). In the same study (p. 5660), C₂H₄ total columns of 2×10¹⁶ molecules cm⁻² were retrieved from IASI observations taken in May 2008, in a fire plume in Eastern Mongolia. It is also worth mentioning that the NN-based retrieval technique used in our study differs largely from the retrieval method applied by Coheur et al. (2009). Moreover, C₂H₄ total columns up to 3.8×10¹⁷ molecules cm⁻² were retrieved during fire events in 2001-2003 southeast Australia from ground-based FTIR observations (Paton- Walsh et al., 2005; Rinsland et al., 2005).

Regarding the background C₂H₄ level around the hotspots, as explained previously, we do not have the possibility to evaluate reliably the IASI measurements in these conditions and therefore cannot rule out that those are potentially biased high. However, we subtract this background C₂H₄ from the column distribution before estimating the emission fluxes (see Methods). If the IASI background C₂H₄

levels were overestimated, this would mean that the emission fluxes we derive underestimate the true emissions and that our IASI-based flux estimates can be seen as conservative.

2. The emissions estimate method is not convincing to me either. The authors use a simple flux method as $E = M/\tau$. There are several issues:

We understand the reservation of the referee on the flux calculation method. However, we are convinced that the simplicity of this method is its strength, as it makes the emission estimates entirely independent of models and of their associated uncertainties which are still large on the C_2H_4 budget (see, e.g., Morgott, 2015; Poisson et al., 2000; Pozzer et al., 2022). Furthermore, such method allows taking full advantage from the high spatial resolution of the supersampling averages. For these reasons, similar mass-balance methods have been heavily favoured in previous state-of-the-art studies to derive emission fluxes from localized anthropogenic point sources of SO_2 (Carn et al., 2007; Fioletov et al., 2011, 2013; Li et al., 2017; McLinden et al., 2016; Wang et al., 2015), NO_2 (Beirle et al., 2011; Ghude et al., 2013; Liu et al., 2016), $HCHO$ (Zhu et al., 2014) and NH_3 (Van Damme et al., 2018). Similar mass-balance approaches are also used to calculate methane fluxes from super-emitters (Duren et al., 2019; Frankenberg et al., 2016; Jongaramrungruang et al., 2019; Varon et al., 2018, 2021). Therefore, as demonstrated in the replies below, we are confident that the method is robust, relies on unique observations and supports the major conclusions of this study.

1) I'm not convinced whether using a constant chemical lifetime for C_2H_4 is right. As the authors mentioned, the chemical lifetime of C_2H_4 depends on O_3 and OH level, which should vary significantly in different regions.

We agree with the referee that using a constant chemical lifetime of C_2H_4 is a simplification. Nevertheless, accounting for a changing chemical lifetime of the target gas based on the concentration in oxidants would represent a real challenge because of the lack of global and consistent constraints. Indeed, as detailed in the reply to point 4) here below, global models would be of little use for such an endeavour because of their too coarse horizontal resolution relative to the spatial extent of the C_2H_4 point sources (i.e., a typical hotspot is "diluted" in a model grid cell).

To account for uncertainties and variations of the C_2H_4 chemical lifetime, we also calculated the emission fluxes from the IASI measurements by assuming two "extreme" C_2H_4 lifetimes of 2 and 24 h, as explained in the manuscript. These changes of lifetime value propagate linearly on the uncertainty in the flux estimate, and the satellite-based fluxes that were obtained with these values can be seen, respectively, as conservative upper- and lower-bound estimates of C_2H_4 emissions. Within these bounds, the EDGAR emissions underestimate by 1-2 order(s) of magnitude the industrial fluxes of C_2H_4 as derived from IASI, providing robust support for the conclusions of the paper.

A major step forward would consist in estimating the atmospheric residence time of C_2H_4 for each point source, based on the satellite measurements themselves (e.g., based on the extent of the pollution plume as done for SO_2 in Fioletov et al., 2015, 2016). However, as this effort requires very accurate satellite measurements, it is currently not feasible due to the uncertainties on the IASI retrievals of background C_2H_4 . With the future launch of the IASI New Generation instrument (IASI-NG), which will benefit from much higher instrumental performance compared to its predecessor, such a method could potentially be workable.

2) It's not clear to me why the authors didn't consider transport. For species like C_2H_4

that
are relatively longer-lived, the loss against dispersion should be much larger than the
loss due

to chemistry. If we assume 10m/s wind speed, for the length scale of 20 km, the lifetime of C₂H₄ against dispersion is 2.7 hours, which is much shorter than the chemical lifetime of 12 hours. It's confusing that the authors rotate the observations downwind, but never accounted for wind speed in the estimates of emissions.

By combining the wind-rotated super-resolution and the mass-balance approach (under steady state) to estimate emission fluxes from the point-sources, we did consider the transport of C₂H₄ (and we do not need to account for the wind speed). Indeed, the emission fluxes are obtained from the supersampled downwind average by calculating the total mass of C₂H₄ above the source and in the transport plume. To that end, as shown in Suppl. Figs 6-8, we delimit a box around the point-source, which the size is adjusted to encompass the hotspot and the whole transport plume. In most cases, the box limits extend in the range of ~50 km from the point-source (hence beyond the 20-30 km length scale of the hotspot). This way, the C₂H₄ amount that would be transported out of that box can be considered to be negligible. Please also note that both wind rotation and super-resolution preserve the overall mass of C₂H₄ around the point source, as demonstrated by Clarisse et al. (2019). As emphasized in the Methods, we also assume that the C₂H₄ mass in that box is at steady state, hence has an effective lifetime that can be regarded as the mean lifetime of the different parts of this mass. Then, as illustrated in Suppl. Figs 6-8, a background C₂H₄ level (calculated specifically for each point source) is subtracted from the downwind column distribution. As a result, what is left inside the adjusted box is the C₂H₄ mass emitted by the point source (panel b in Suppl. Figs 6- 8), which is eventually used to derive the C₂H₄ emission fluxes. As explained above, this mass- balance approach is widely recognised as robust to derive localized anthropogenic point- sources of short-lived pollutants (see the references given above).

3) The assumption of 12 hour lifetime is not correct to me looking at Figure S3. It seems the the length scale of C₂H₄ plume is around 20 km. If the lifetime is 12 hours, shouldn't it transport much further? Why is it just concentrated near source?

We agree with the referee that, based on the extent of the hotspot in Suppl. Figs 1-3 and 6-8, and on the referee's estimates of the C₂H₄ lifetime against dispersion, the atmospheric residential time of C₂H₄ is likely shorter than 12 h in the vicinity of anthropogenic point- sources. This is consistent with the C₂H₄ lifetimes of a few hours derived from aircraft measurements in industrial plumes (e.g., Cho et al., 2021; Washenfelder et al., 2010; Wert et al., 2003), and likely attributed to the elevated concentration in atmospheric oxidants in such polluted areas. Nonetheless, we assumed in our work a longer lifetime (12 h) for three reasons: 1) by potentially overestimating the lifetime, we remain on the conservative side on the magnitude of the emission fluxes, allowing to draw robust conclusions when comparing to the bottom-up inventory 2) Although many plumes exhibit indeed a 20-30 km spatial extent, we cannot rule out that the real plumes are longer but cannot be distinguished by IASI from the relatively noisy background around the point-source. 3) The atmospheric residential lifetime of C₂H₄ can be longer than only a few hours at seasons where the atmospheric chemistry is less efficient and at places where the level of oxidants is lower. A conservative 12-h lifetime of C₂H₄ is therefore meant at including these possibilities. As demonstrated by our flux calculations assuming a lifetime of only 2 h, a shorter lifetime yields higher IASI emission fluxes from the point sources and increases even more the gap between the satellite- based and EDGAR industrial emissions, which does not change the conclusions of our study.

4) I think a better way to compare satellite-based C₂H₄ with EDGAR emission inventory is to run chemical transport models to compare model estimated C₂H₄ columns vs. satellite retrieved columns, which can avoid the issues with the emission estimates methods.

In theory, we agree of course that the IASI-C₂H₄ measurements could be compared with global models driven by emission inventories. However, at the spatial scale at which we determine the emission fluxes from IASI, it is not advantageous since current state-of-the-art global models do not run at a spatial resolution high enough (1° × 1° in the best case) to exploit the IASI-derived hyper-resolved map of C₂H₄ (0.01° × 0.01°). Considering the limited spatial extent of most of the C₂H₄ point sources (usually 20-30 km), the hotspot emissions would be significantly diluted within the model grid cells at much coarser resolution and the benefit of the hyper-resolution would be completely lost. Moreover, as stressed above, the C₂H₄ budget from global models is affected by substantial uncertainties. There is indeed no consensus as to the role and amplitude of the different sources and sinks of atmospheric C₂H₄ (Morgott et al., 2015; Poisson et al., 2000; Pozzer et al., 2022). For example, estimates of the anthropogenic sources range from 1.05 to 9.20 Tg yr⁻¹, the biogenic sources from 2.7 to 26.2 Tg yr⁻¹, and the contribution of biomass burning from 3.55 to 5.35 Tg yr⁻¹ (Poisson et al., 2000; Pozzer et al., 2022; Sawada and Totsuka, 1986). Considering this, in a satellite-to-model comparison it would be extremely complex to disentangle the discrepancies due to the satellite vs. EDGAR emission fluxes, from those caused by the disparities and deficiencies between models. These reasons justify in our opinion the use of the simple, more robust approach as the one implemented in our study.

3. While it's interesting to provide high resolution estimates of C₂H₄, it's not clear to me what are the values of high-resolution data. The authors actually didn't take advantage of the spatial patterns of C₂H₄ in the emission estimates, and they simply take the spatial average of C₂H₄ of a big box. If so, why do they even need high spatial resolution?

The high resolution is first needed for the detection and identification of the C₂H₄ point-sources. As shown in Suppl. Fig. 1 for an extended C₂H₄ hotspot, the oversampling techniques yield a much better representation of the point-sources. It is then easier to identify the potential emitter with the help of satellite imagery. For instance, Clarisse et al. (2019) demonstrated that the wind-rotated supersampling technique improved performance in the geo-allocation of NH₃ point-sources by reducing the median detection distance from 3.9 km with a regular oversampling, to 1.5 km. It is also worth mentioning that most C₂H₄ point-sources have a limited spatial extent in the range of 20-30 km and are therefore merely undetectable with a regular binned average of satellite data. This is illustrated in the figure here below, similar to Suppl. Fig. 1, but for a point-source associated with a petrochemical cluster at Novopolotsk (Belarus). This new figure has been added to the Supplementary Information as Fig. 2. The advantage of the super-resolution over a regular oversampling has also been illustrated by Clarisse et al. (2019), who doubled the number of NH₃ point-sources detected with IASI (>500 vs. 248) compared to the work of Van Damme et al. (2018) in which a regular oversampling was used. Moreover, the high resolution is key to obtain a compliant representation of the C₂H₄ transport plume. As explained in the replies above, knowing the extent of that plume is required to adjust the size of the box used to calculate the emissions from the point-source. Finally, the super-resolution reproduces more realistically the strength of the hotspot (and transport plume) which contrasts much more with the background, background that is then evaluated and subtracted from the column distribution to calculate the emission fluxes.

Supplementary Fig. 2 | a, Binned average on a $0.15^\circ \times 0.15^\circ$ spatial resolution grid, b, oversampled average, c, wind-rotated oversampling, and d, wind-rotated supersampling of the IASI C_2H_4 HRI at a $0.01^\circ \times 0.01^\circ$ spatial resolution, around the hotspot of Novopolotsk (Belarus). The coordinates of the presumed emitter, marked by a white square, is used as the wind rotation centre. The area delimited by the black dashed line is used to calculate the averaged downwind HRI value. This is a typical example of a C_2H_4 hotspot that would be difficult to find without the wind-rotated supersampling.

4. The results section are overall qualitative. Figure 2 only shows the distribution of the hotspots, but what the concentration levels? How do they vary spatially? I actually found Figure S2 more informative. Especially since the next section only focused on 53 out of 300 hotspots, there is almost no quantitative information about the other hotspots.

We agree that Suppl. Fig. 2 is very informative for the readers, therefore we now include it in the main text as the new Fig. 1. Please note that we have increased the colour contrast between the IASI data and the background satellite imagery.

[Redacted]

The first part of this study is indeed overall qualitative, but that is exactly what this work is meant for: demonstrating for the first time that a large number of C_2H_4 anthropogenic point-sources worldwide can be tracked down from space and that the source emitters are related to heavy industries and megacities. Considering that C_2H_4 is a primary compound directly emitted (conversely to UV-Vis VOC targets formaldehyde and glyoxal), a high-yield precursor of tropospheric O_3 , and that many other short-lived (sometimes harmful) VOCs are emitted conjointly (e.g., propylene, benzene, toluene, xylene), we firmly believe that the present work is an important milestone to comprehend the impact of industrial gas emissions and that it will motivate future efforts to monitor from space VOC emissions from industrial sources. Regarding the emission fluxes from a selection of hotspots, as quoted by referee #3, we “*are really pushing the data to its limits*”. These results have indeed been obtained by applying the most advanced data retrieval and spatial sampling techniques to date. For instance, the wind-rotated supersampling has been developed in our group and so far, applied only to IASI- NH_3 data (Clarisse et al., 2019). This study should be seen as a first proof of concept with IASI and should also be seen in the light of future satellite missions such as IASI-NG, that will undoubtedly allow much tighter constraints on the emission fluxes from point-sources.

Minor comments:

Figure 1: Better label the location of the sources on the Google Earth images.

We are not sure to understand what the reviewer means here. The side panels offer close-up views of the point sources, on a scale of 1×1 km, and centred close to where the maximum HRI is found. Within this box, it is not possible for us to point more precisely at the building that is responsible for the C_2H_4 emissions. It is also likely the case that several nearby sources (e.g., different chimneys) contribute to the same hotspot.

Figure 3: Table S2 suggests large variation of IASI flux with lifetime, but the uncertainty of the C_2H_4 flux is not shown here. I'd suggest include an uncertainty estimates here.

Uncertainties based on the lifetime are difficult to add as error bars to the histogram in Fig. 3a because changing the lifetime in the flux estimates redistributes the population of hotspots amid the different categories of IASI-to-EDGAR ratio. For example, a longer lifetime (24h) increases the weight of the low

IASI-to-EDGAR ratios, while it decreases the weights of the high ratios. Therefore, we propose instead to add to the Supplementary Information a new figure (here below) with three histogram charts similar to Fig. 3a but reporting the IASI-to-EDGAR ratios assuming a C_2H_4 lifetime of, respectively, 24, 12 and 2h. In this new figure, the panel b corresponds to Fig. 3a, but with a different y-axis.

Supplementary Fig. 9 | Comparison between IASI-derived and EDGAR C_2H_4 fluxes. a, Ratio between the IASI-derived and EDGAR v4.3.2 C_2H_4 emissions, over 57 selected hotspots. EDGAR emissions were computed, respectively, for all sectors (blue) and the industrial sectors only (red). The IASI-derived fluxes were calculated assuming a C_2H_4 lifetime of 24h. b, c, Same as a, but with a C_2H_4 lifetime of 12 and 2h, respectively.

Figure S3: It's not clear which is the downwind direction.

In this work, when the wind rotation is applied, the winds are aligned to the east (in the x direction). We have added this information to the figure caption:

“a, c, Distribution of C_2H_4 HRI and b, d, columns obtained with the wind-rotated supersampling over two example hotspots. The white squares give the location of the presumed emitters used as the rotation points. In this work, the winds are realigned to the east (in the x direction). The column distributions exhibit a noisier background compared with the HRI.”

References:

Coheur, P.-F., Clarisse, L., Turquety, S., Hurtmans, D., and Clerbaux, C.: IASI measurements of reactive trace species in biomass burning plumes, *Atmos. Chem. Phys.*, 9, 5655–5667, <https://doi.org/10.5194/acp-9-5655-2009>, 2009.

Reviewer #3 (Remarks to the Author):

This article presents ethylene observations made from the IASI instruments on the Metop satellites. Using IASI and advances in retrieval algorithms and long-term averaging, the authors build a high-resolution global dataset of ethylene at 0.01x0.1 degree resolution (~1km) averaged for 2008-2020. This is done using a wind-adjust super-resolution oversampling, neural networks, and a recently developed “whitening” spectral technique (all detailed thoroughly in the text). I commend the authors for their analysis. They are really pushing the data to its limits.

Using this dataset, they are able to detect 336 ethylene emission global hotspots. To my knowledge, this is the first time industrial emissions of ethylene have been detailed from space (there has been previous detection in biomass burning plumes) and the number they have detected is impressive. Ethylene flux is quantified for 57 large hotspots. Of significance, the authors find the widely-used emissions database EDGAR underestimates ethylene often by 1-2 orders of magnitude and completely misses 35 of 53 industrial sources.

The paper shows a large range of fluxes assuming different lifetimes (see Table 2 in the Supplement) and points to the need to quantify lifetimes of ethylene for various conditions, and the need to improve speciation knowledge of VOC emissions in inventories. As a result, there is still a lot of work to do in field measurements and modeling to make this data really useful for top-down (satellite-based) emissions inventories. Overall the mapping of ethylene hotspots is an interesting development in and of itself, but the results also point to some major uncertainties in our current inventories and understanding of global VOC emissions.

The paper presents exciting results which will be of significant interest to several communities in atmospheric science, including those working on trace gas retrievals from space and atmospheric modellers. It will also be of significant interest to regulatory agencies who depend on accurate emissions inventories, and to the general public and policy makers who are interested in pin-pointing sources of industrial pollutants. I would recommend this work be published in any journal and I believe it of high quality and broad enough interest for Nature Communications.

We thank the reviewer for the positive evaluation of the paper and for the constructive comments that really helped improving the manuscript. Please find in blue here below our answers to the reviewer’s comments and the changes made to the manuscript.

Specific Comments:

Line 36: Also early detection by Alvarado et al. (2011) using TES: <https://doi.org/10.3390/atmos2040633>

We thank the referee for this reference. We have added it to the manuscript.

Line 72: Can you be specific here about online sources? Is this all done by visual interpretation and then looking up potential sources? It’s a big vague but I’m guessing this is how it’s done?

Indeed, it is mainly based on visual interpretation. With the help of satellite visible imagery, we tried to identify the presence, type, and location of potential emitters of C₂H₄ (e.g., coal-fired power plant, petrochemical refinery...) in the vicinity of the centre of the hotspot detected by IASI. Then, we collected information online (e.g., website of the companies) to identify the industries and their activities (e.g., the presence of an ethylene cracker in a petrochemical cluster). All this information was gathered for each hotspot in a vast table, based on which we finally attributed the different categories to the point-sources.

In the manuscript, we have clarified the sentence as follows:

“With the help of satellite visible imagery similar to what we illustrate with Fig. 1, and information collected online (e.g., on the companies and type of activities), we pinpoint for each hotspot the likely emitters of ethylene and show that in most cases they belong to three specific types of industry.”

Line 321: It's not clear to me until much later in text... Are you using merged data from all these three IASI instruments? Are they consistent with each other during overlapping periods? Spatial resolution of IASI is mentioned later but I think it should also be introduced here with other instrument information.

We indeed used the merged data from the three IASI instruments, as those are identical, stable, and very well calibrated. Their corresponding C₂H₄ datasets show an excellent agreement during their overlapping periods. We have now added this information, as well as the typical spatial resolution of IASI, in this part of the text as follows:

“The measurements of ethylene (C₂H₄) are derived from the hyperspectral observations recorded by the Infrared Atmospheric Sounding Interferometer (IASI), embarked on the three sun-synchronous, polar-orbiting meteorological satellite platforms Metop²². IASI/Metop-A, -B, and -C are providing data since, respectively, October 2007, March 2013, and September 2019 (IASI/Metop-A was decommissioned in late 2021). In this work, the three IASI C₂H₄ datasets are used together and show an excellent agreement during their overlapping periods. IASI is a Fourier transform spectrometer with an apodized spectral resolution of 0.5 cm⁻¹ (spectrally sampled at 0.25 cm⁻¹), which measures in a nadir geometry the radiance of the Earth and of the atmosphere in the thermal infrared spectral range between 645 and 2760 cm⁻¹ without gap²². The radiometric noise in the spectral range of the main C₂H₄ absorption feature near 949 cm⁻¹ is ~0.15 K for a reference blackbody at 280 K. One IASI instrument provides near global coverage twice a day, with measurements at ~09:30 am and pm (local equator crossing time). At nadir, the footprint of an IASI measurement is a 12 km diameter circle, and at off-nadir angles, an ellipse elongated up to 20 × 39 km.”

Line 364: Probably should refer to a figure here to give evidence of this statement. We now refer to the Supplementary Fig. 5:

“Further, we present an unambiguous detection of ethylene in IASI spectra over C₂H₄ enhancements (see also Supplementary Fig. 5).”

Line 384: Is the peak concentration fixed at the surface everywhere, or does it vary away from a priori sources? I imagine it's the first as this is in the training set development but can you please clarify here.

In this study, the peak concentration is indeed fixed at the surface everywhere. In the Methods, we have cleared up the text as follows:

“with \square_0 the peak height of the vertical profile (in km) as a measure of the altitude of the bulk of C₂H₄,

\square the standard deviation of the Gaussian function as a measure of the thickness of the C₂H₄ layer (in km; assigned as explained below), and \square (in ppb) a scaling factor of the profile that controls $\text{vmr}_{(\text{C}_2\text{H}_4)}$ at \square_0 and hence the C₂H₄ abundance in the forward simulations. In this work, the peak concentration was always fixed at the surface to be representative for observations over hotspots, where the bulk of C₂H₄ is assumed to be close to the surface.

[...]

The value of \square was set to the ERA5 boundary layer height.”

Line 445: Presumably there is quite a lot of temporal variability in 13 years that you don't detect. How does this affect the estimates and super resolution calculation? For instance, emitters have been changing rapidly in a place like China. Could there be large uncertainties from plants coming on and offline? How is this dealt with the EDGAR comparisons?

The values of C₂H₄ HRI and top-down emissions that we report here are averages over the entire IASI time series (i.e., 2008-2020). We acknowledge that this is a limitation of the study and that this introduces some uncertainties. For example, for industries that started their activities in the middle of the IASI time series, the satellite averages overestimate their emissions for the first half of the period and underestimate their emissions for the second half. Also, industries that opened after 2012 are missing in the EDGAR speciated VOC emissions. In the revision, we have now investigated whether the IASI data can be used to reveal temporal changes of C₂H₄ hotspots. For this, we applied the wind-rotated supersampling to the IASI data over 2008-2011, 2012-2014, 2015-2017 and 2018-2020. Analysis of the multi-year high-resolution maps reveals that the representation of many C₂H₄ point-sources, especially the weak hotspots, is significantly more difficult because of fewer satellite data and, as a result, of lower signal-to-noise ratios. Nonetheless, for a series of major hotspots, the representation of the point-sources remains good enough to allow a temporal assessment. Interestingly, some of them exhibit a significant increase over time of the C₂H₄ HRI values and associated top-down emissions. Two examples are displayed in the figures below: 1) the Hainan point-source (Inner Mongolia, China) concentrating numerous coal-related activities, and 2) the Yanbu petrochemical hub (Saudi Arabia). In both cases, the increasing top-down C₂H₄ fluxes are corroborated by a densification of industrial complexes seen in satellite imagery taken within each period. We added the first figure to the manuscript (and the second to the Supplementary Information) and added a discussion of the temporal aspects as reported here below.

[Redacted]

fluxes calculated for each time period are provided in black. **Bottom panels**, Zoom-ins on the presumed emitter(s) with satellite visible imagery taken within each time period.

[Redacted]

“To investigate whether IASI can reveal temporal changes in the anthropogenic emissions of atmospheric C₂H₄, we applied the wind-adjusted super-resolution technique to the satellite data over four different time periods: 2008-2011, 2012-2014, 2015-2017 and 2018-2020. Despite the decrease in the signal-to-noise ratio, the representation of the point-sources and transport plumes is good enough to allow studying the time evolution of the largest hotspots. For most, no significant trends were observed, but for others, the hyperfine resolution maps reveal a clear progressive enhancement of the downwind C₂H₄ HRI average with time, as illustrated in Fig. 5 for a coal-related hotspot in Inner Mongolia, Central China. Another example over a large petrochemical hub in Saudi Arabia is presented in Supplementary Fig. 10. For these two examples, the top-down C₂H₄ emissions grow respectively from 7.2×10⁻² and 8.8×10⁻² kg s⁻¹ for the 2008-2011 period, to 2.6×10⁻¹ and 2.9×10⁻¹ kg s⁻¹ over 2018-2020. Satellite visible imagery supports this finding, as new industrial complexes are seen to appear over the different periods (Fig. 5, Supplementary Fig. 10). Similar examples are found in the central and western regions of China, which have recently undergone programs of industrial development that rely heavily on the exploitation of local coal resources as raw materials and for energy production^{25, 26}. Currently, the temporal assessment is limited to the most prominent point-sources and to multiyear time blocks. These examples, however, represent the first demonstration of how infrared sounders can be used to monitor industrial emissions of VOCs, a capability that is surely going to improve with the next-generation infrared sounders that will offer better spectral resolution and lower noise in case of IASI- NG³³, and increased temporal sampling in case of the geostationary infrared sounder onboard the Meteosat Third Generation (MTG-IRS) satellite (<https://www.eumetsat.int/meteosat-third-generation>).”

Line 470: “*all the uncertainties associated with the retrieved column*”: Perhaps these uncertainties have been previously discussed, but feel like I need to be reminded here of what they are. Also here or elsewhere, can you say a few more words about quantitative uncertainties? What are the errors introduced by the vertical sensitivity of IASI?

Following the ANNI v3 retrieval approach, an uncertainty on each retrieved column is evaluated by propagating through the NN the uncertainties of the different input variables of the NN (Whitburn et al., 2016; Franco et al., 2018). For example, all the layers of the temperature profile are independently perturbed by a random noise up to 1 K, and each HRI value is perturbed with a Gaussian noise characterized by a mean of zero and a standard deviation of one. Then we calculated the relative errors of the perturbed NN output over the actual retrieved columns. The column uncertainties depend primarily on the gas abundance and thermal contrast. The typical uncertainty on an individual retrieved column is evaluated to be below 50% for columns above 1×10^{16} molecules cm^{-2} and positive surface-atmosphere thermal contrasts. Consistent with the other VOCs retrieved from IASI (e.g., Franco et al., 2018, 2020), the error values increase beyond 50% for lower columns as the weak concentrations approach the IASI detection threshold, and for zero or negative thermal contrasts which reduce the IASI sensitivity to C_2H_4 . However, these uncertainties are significantly reduced for the averages calculated here, due to the large number of measurements averaged per grid cell.

We have added this information to the Methods, in the “*C₂H₄ retrieval and IASI dataset*” section:

“An uncertainty on each retrieved column is evaluated by propagating the uncertainties of the different input variables of the NN, as detailed by Refs 34, 36, 38. The typical uncertainty on an individual retrieved C₂H₄ column is below 50% for columns above 1×10^{16} molecules cm^{-2} and positive surface- atmosphere thermal contrasts. Consistent with the other VOCs retrieved from IASI³⁶⁻³⁸, the error values increase beyond 50% for lower columns as the weak C₂H₄ concentrations approach the IASI detection threshold, and for weak or negative thermal contrasts which reduce the IASI sensitivity. However, these uncertainties are significantly reduced for the column averages calculated here, due to the large number of measurements averaged per grid cell (see below).”

To assess the errors on the vertical sensitivity of IASI, we built two C_2H_4 vertical profiles with the equation R1 in the Methods, one with the peak at surface and the other with the peak at 0.5 km altitude. In both cases, we assumed a C_2H_4 vmr of 5 ppb at the peak altitude. Using the 1976 US standard atmosphere, we calculated corresponding C_2H_4 total columns of 2.13×10^{16} and 2.64×10^{16} molecules cm^{-2} , respectively. This represents roughly a 20% difference, which is typically well below the 1-2 order(s) of magnitude of difference that we observe between the IASI-derived and the EDGAR emission fluxes over the industrial point-sources. We have also added this result to the Methods:

“To evaluate the error on the IASI vertical profile, we used R1 to build two C₂H₄ vmr profiles assuming a peak concentration of 5 ppb ($\text{vmr}_{(\text{C}_2\text{H}_4)}$ at z_0) at, respectively, surface and 0.5 km altitude. Using the 1976 US standard atmosphere, we calculated C₂H₄ total columns of 2.13×10^{16} and 2.64×10^{16} molecules cm^{-2} from these two profiles. This 20% difference is well below the typical 1-2 order(s) of magnitude of discrepancy that we observe between the top-down and EDGAR emissions over industrial point- sources.”

Line 575: What is the background ethylene typically removed? How different is this from modelled background ethylene?

The C_2H_4 column background that is removed depends on the location of the hotspot. Typically, in a highly polluted regions like North-East China, the background column ranges between 5 and 10×10^{15} molecules cm^{-2} , while it usually falls below $2-3 \times 10^{15}$ molecules cm^{-2} for more isolated hotspots or in less polluted areas. It is difficult to evaluate the satellite

background with model simulations, since the

estimates of the different C₂H₄ sources diverge substantially between models (see, e.g., Morgott, 2015; Poisson et al., 2000; Pozzer et al., 2022). Nevertheless, Toon et al. (2018) reported C₂H₄ total columns, derived from ground-based FTIR measurements, usually below 2-3×10¹⁵ molecules cm⁻² in remote regions, while column amounts up to 20×10¹⁵ molecules cm⁻² were measured at urban sites in California. The IASI data are relatively consistent with these ground-based measurements.

Supp. Fig 1: Is wind-rotated oversampling and super-sampling showing all ethylene to the east because

this direction is considered “downwind”? Might be useful to note this somewhere.

The wind rotation realigns, on a daily basis, each satellite measurement around the presumed gas emitter according to the daily horizontal wind direction. Applied to a satellite time series, this yields a distribution of the measurements in which the winds blow in the same direction, arbitrarily chosen, from the point-source. Here, the winds are aligned to the east (in the x direction). We have added this information to the figure caption:

“a, Binned average on a 0.15° × 0.15° spatial resolution grid, b, oversampled average, c, wind-rotated oversampling, and d, wind-rotated supersampling of the IASI C₂H₄ HRI at a 0.01° × 0.01° spatial resolution, around the hotspot of Mengxi Park (Inner Mongolia, China). The coordinates of the presumed emitter, marked by a white square, is used as the wind rotation centre. In this work, the winds are realigned to the east (in the x direction). The area delimited by the black dashed line is used to calculate the averaged downwind HRI value.”

Supp. Fig 2: I find the color scheme superimposed on the global visible imagery difficult to read (for example, shades of green on green, and is that white on a dark blue ocean? – I can't tell). I find it fine to look at in the very high resolution imagery. However, there are less contrasts in the background of the global imagery which is maybe why it is difficult to see what is ethylene and what is the surface.

We have improved the visibility in that figure by increasing the colour contrast between the IASI data and the background satellite visible imagery. We have also changed the markers giving the location of hotspots and areas that undergo a thorough spectral analysis. Please also note that this figure appears now in the main manuscript as the new Fig. 1.

[Redacted]

Technical Comments:

We thank the referee for all his/her corrections that improve the manuscript.

Line 14: Awkward sentence, suggest change to “Here, we track from space over 300 worldwide hotspots of ethylene, the most abundant industrially produced organic compound.”

We have corrected following the referee’s suggestion.

Line 31: Similar awkward sentence, “which contributes” refers to ethylene and not “sources” so needs to come earlier after the sentence subject, or change to “and contributes”.

We have changed to “*and contributes*”.

Line 35: Same sentence structure issue, “dominated by natural sources” does not refer to troposphere but to “concentration”

The sentence now reads:

“Although locally it can be emitted in vast quantities by biomass burning^{10, 11}, its background concentration is dominated by natural sources and remains mostly below 0.1 part per billion (ppb) in the global troposphere¹²⁻¹⁴.”

Line 52: “We took benefit of the” doesn’t make sense and sentence is a bit awkward.

We have changed to:

“To take advantage of the extensive IASI time series, we applied a wind-adjusted super-resolution technique to the HRI dataset²³, which allows increasing the spatial resolution of satellite data beyond the native resolution of the sounder (Methods; Supplementary Fig. 1).”

Line 320: “embarked” not really used in this context. Suggest “flying on”

Corrected.

Line 345: Change to “among which are included the VOCs”

Done.

Line 352: “built in such a way”

Corrected.

Line 425: Change “took benefit of” to “took advantage of”

Done.

Line 568: “In such a way”

Corrected.

References

- Altuzar, V. *et al.* Atmospheric ethene concentrations in Mexico City: Indications of strong diurnal and seasonal dependences. *Atmospheric Environment* **39(29)**, 5219-5225 (2005).
- Alvim, D. S. *et al.* Determining VOCs Reactivity for Ozone Forming Potential in the Megacity of São Paulo. *Aerosol and Air Quality Research* **18(9)**, 2460-2474 (2018).
- An, J. *et al.* Characteristics and source apportionment of VOCs measured in an industrial area of Nanjing, Yangtze River Delta, China. *Atmospheric Environment* **97**, 206-214 (2014).
- Beirle, S. *et al.* Megacity Emissions and Lifetimes of Nitrogen Oxides Probed from Space, *Science* **333(6050)**, 1737-1739 (2011).
- Carn, S. A. *et al.* Sulfur dioxide emissions from Peruvian copper smelters detected by the Ozone Monitoring Instrument. *Geophysical Research Letters* **34(9)**, (2007).
- Chang, C.-C. *et al.* An examination of 7:00-9:00PM ambient air volatile organics in different seasons of Kaohsiung city, southern Taiwan. *Atmospheric Environment* **39(5)**, 867-884 (2005).
- Cho, C. *et al.*, Evolution of formaldehyde (HCHO) in a plume originating from a petrochemical industry and its volatile organic compounds (VOCs) emission rate estimation. *Elementa: Science of the Anthropocene* **9(1)**, (2021).
- Clarisse, L. *et al.* Tracking down global NH₃ point sources with wind-adjusted superresolution. *Atmospheric Measurement Techniques* **12(10)**, 5457-5473 (2019).
- Clerbaux, C. *et al.* Monitoring of atmospheric composition using the thermal infrared IASI/MetOp sounder. *Atmospheric Chemistry and Physics* **9(16)**, 6041-6054 (2009).
- Coheur, P.-F. *et al.* IASI measurements of reactive trace species in biomass burning plumes. *Atmospheric Chemistry and Physics* **9(15)**, 5655-5667 (2009).
- de Gouw, J. A. *et al.* Airborne Measurements of Ethene from Industrial Sources Using Laser Photo- Acoustic Spectroscopy. *Environmental Science & Technology* **43(7)**, 2437-2442 (2009).
- Dominutti, P. A. *et al.* One-year of NMHCs hourly observations in São Paulo megacity: meteorological and traffic emissions effects in a large ethanol burning context. *Atmospheric Environment* **142**, 371- 382 (2016).
- Duren, R. M. *et al.* California's methane super-emitters, *Nature* **575(7781)**, 180-184 (2019).
- Fioletov, V. E. *et al.* Estimation of SO₂ emissions using OMI retrievals. *Geophysical Research Letters* **38(21)**, (2011).
- Fioletov, V. E. *et al.* Application of OMI, SCIAMACHY, and GOME-2 satellite SO₂ retrievals for detection of large emission sources. *Journal of Geophysical Research: Atmospheres* **118(19)**, 11,399-11,418 (2013).
- Fioletov, V. E. *et al.* Lifetimes and emissions of SO₂ from point sources estimated from OMI. *Geophysical Research Letters* **42(6)**, 1969-1976 (2015).
- Fioletov, V. E. *et al.* A global catalogue of large SO₂ sources and emissions derived from the Ozone Monitoring Instrument. *Atmospheric Chemistry and Physics* **16(18)**, 11,497-11,519

(2016).

- Franco, B. *et al.* Spaceborne Measurements of Formic and Acetic Acids: A Global View of the Regional Sources. *Geophysical Research Letters* **47(4)**, e2019GL086239 (2020).
- Frankenberg, C. *et al.* Airborne methane remote measurements reveal heavy-tail flux distribution in Four Corners region. *Proceedings of the National Academy of Sciences* **113(35)**, 9734-9739 (2016).
- Ghude, S. D. *et al.* Application of satellite observations for identifying regions of dominant sources of nitrogen oxides over the Indian Subcontinent. *Journal of Geophysical Research: Atmospheres* **118(2)**, 1075-1089 (2013).
- Huang, G. *et al.* Speciation of anthropogenic emissions of non-methane volatile organic compounds: a global gridded data set for 1970—2012. *Atmospheric Chemistry and Physics* **17(12)**, 7683-7701 (2017).
- Jia, C. *et al.* Non-methane hydrocarbons (NMHCs) and their contribution to ozone formation potential in a petrochemical industrialized city, Northwest China. *Atmospheric Research* **169**, 225-236 (2016).
- Johansson, J. K. E. *et al.* Emission measurements of alkenes, alkanes, SO₂, and NO₂ from stationary sources in Southeast Texas over a 5 year period using SOF and mobile DOAS. *Journal of Geophysical Research: Atmospheres* **119(4)**, 1973-1991 (2014).
- Jongaramrungruang, S. *et al.* Towards accurate methane point-source quantification from high-resolution 2-D plume imagery. *Atmospheric Measurement Techniques* **12(12)**, 6667-6681 (2019).
- Li, C. *et al.* India Is Overtaking China as the World's Largest Emitter of Anthropogenic Sulfur Dioxide, *Scientific Reports* **7(1)**, (2017).
- Liu, F. *et al.* NO_x lifetimes and emissions of cities and power plants in polluted background estimated by satellite observations. *Atmospheric Chemistry and Physics* **16(8)**, 5283-5298 (2016).
- McDuffie, E. E. *et al.* A global anthropogenic emission inventory of atmospheric pollutants from sector- and fuel-specific sources (1970-2017): an application of the Community Emissions Data System (CEDS). *Earth System Science Data* **12(4)**, 3413-3442 (2020).
- McLinden, C. A. *et al.* Space-based detection of missing sulfur dioxide sources of global air pollution. *Nature Geoscience* **9(7)**, 496-500 (2016).
- Morgott, D. A. Anthropogenic and biogenic sources of Ethylene and the potential for human exposure: A literature review. *Chemico-Biological Interactions* **241**, 10-22 (2015).
- Na, K. *et al.* Concentrations of volatile organic compounds in an industrial area of Korea. *Atmospheric Environment* **35(15)**, 2747-2756 (2001).
- Paton-Walsh, C. *et al.* Measurements of trace gas emissions from Australian forest fires and correlations with coincident measurements of aerosol optical depth. *Journal of Geophysical Research: Atmospheres* **110(D24)**, (2005).
- Poisson, N. *et al.* Impact of Non-Methane Hydrocarbons on Tropospheric Chemistry and the Oxidizing Power of the Global Troposphere: 3-Dimensional Modelling Results. *Journal of Atmospheric Chemistry* **36(2)**, 157-230 (2000).
- Pozzer, A. *et al.* Simulation of organics in the atmosphere: evaluation of EMACv2.54 with the

Mainz Organic Mechanism (MOM) coupled to the ORACLE (v1.0) submodel. *Geoscientific Model Development* **15(6)**, 2673-2710 (2022).

- Rinsland, C. P. *et al.* High spectral resolution solar absorption measurements of ethylene (C₂H₄) in a forest fire smoke plume using HITRAN parameters: Tropospheric vertical profile retrieval. *Journal of Quantitative Spectroscopy and Radiative Transfer* **96(2)**, 301-309 (2005).
- Ryerson, T. B. *et al.* Effect of petrochemical industrial emissions of reactive alkenes and NO_x tropospheric ozone formation in Houston, Texas. *Journal of Geophysical Research* **108(D8)**, (2003).
- Sawada, S. and Totsuka, T. Natural and anthropogenic sources and fate of atmospheric ethylene. *Atmospheric Environment* **20(5)**, 821-832 (1986).
- Shen, L. *et al.* Sources Profiles of Volatile Organic Compounds (VOCs) Measured in a Typical Industrial Process in Wuhan, Central China. *Atmosphere* **9(8)**, 297 (2018).
- Simpson, I. J. *et al.* Characterization, sources and reactivity of volatile organic compounds (VOCs) in Seoul and surrounding regions during KORUS-AQ. *Elementa: Science of the Anthropocene* **8**, (2020).
- Stohl, A. *et al.* Evaluating the climate and air quality impacts of short-lived pollutants. *Atmospheric Chemistry and Physics* **15(18)**, 10529-10566 (2015).
- Tiwari, V. *et al.* Ambient levels of volatile organic compounds in the vicinity of petrochemical industrial area of Yokohama, Japan. *Air Quality, Atmosphere & Health* **3(2)**, 65-75 (2009).
- Toon, G. C. *et al.* Measurements of atmospheric ethene by solar absorption FTIR spectrometry. *Atmospheric Chemistry and Physics* **18(7)**, 5075-5088 (2018).
- Van Damme, M. *et al.* Industrial and agricultural ammonia point sources exposed. *Nature* **564(7734)**, 99-103 (2018).
- Van Damme, M. *et al.* Global, regional and national trends of atmospheric ammonia derived from a decadal (2008-2018) satellite record. *Environmental Research Letters* **16(5)**, 055017 (2021).
- Vander Auwera, J. *et al.* Self-broadening coefficients and improved line intensities for the v₇ band of ethylene near 10.5 μm, and impact on ethylene retrievals from Jungfraujoch solar spectra. *Journal of Quantitative Spectroscopy and Radiative Transfer* **148**, 177-185 (2014).
- Varon, D. J. *et al.* Quantifying methane point sources from fine-scale satellite observations of atmospheric methane plumes. *Atmospheric Measurement Techniques* **11(10)**, 5673-5686 (2018).
- Varon, D. J. *et al.* High-frequency monitoring of anomalous methane point sources with multispectral Sentinel-2 satellite observations. *Atmospheric Measurement Techniques* **14(4)**, 2771-2785 (2021).
- Velasco, E. *et al.* Distribution, magnitudes, reactivities, ratios and diurnal patterns of volatile organic compounds in the Valley of Mexico during the MCMA 2002 & 2003 field campaigns. *Atmospheric Chemistry and Physics* **7(2)**, 329-353 (2007).
- Walker, J. C. *et al.* An effective method for the detection of trace species demonstrated using the MetOp Infrared Atmospheric Sounding Interferometer. *Atmospheric Measurement Techniques* **4(8)**, 1567-1580 (2011).
- Wang, S. *et al.* Satellite measurements oversee China's sulfur dioxide emission reductions

from coal- fired power plants. *Environmental Research Letters* **10(11)**, 114015 (2015).

Washenfelder, R. A. *et al.* Characterization of NO_x, SO₂, ethene, and propene from industrial emission sources in Houston, Texas. *Journal of Geophysical Research* **115(D16)**, (2010).

Wei, W. *et al.* Characterizing ozone pollution in a petrochemical industrial area in Beijing, China: a case study using a chemical reaction model. *Environmental Monitoring and Assessment* **187(6)**, (2015).

Wei, W. *et al.* Speciated VOCs emission estimate for a typical petrochemical manufacturing plant in China using inverse-dispersion calculation method. *Environmental Monitoring and Assessment* **190(8)**, (2018).

Wert, B. P. *et al.* Signatures of terminal alkene oxidation in airborne formaldehyde measurements during TexAQS 2000. *Journal of Geophysical Research: Atmospheres* **108(D3)**, 4104 (2003).

Whitburn, S. *et al.* A flexible and robust neural network IASI-NH₃ retrieval algorithm. *Journal of Geophysical Research: Atmospheres* **121(11)**, 6581-6599 (2016).

Wu, Y. *et al.* The Characteristics of Ambient Non-Methane Hydrocarbons (NMHCs) in Lanzhou, China. *Atmosphere* **10(12)**, 745 (2019).

Zhang, X. *et al.* Volatile Organic Compounds in a Petrochemical Region in Arid of NW China: Chemical Reactivity and Source Apportionment. *Atmosphere* **10(11)**, 641 (2019).

Zheng, H. *et al.* Compositions, sources and health risks of ambient volatile organic compounds (VOCs) at a petrochemical industrial park along the Yangtze River. *Science of The Total Environment* **703**, 135505 (2020).

Zhu, L. *et al.* Anthropogenic emissions of highly reactive volatile organic compounds in eastern Texas inferred from oversampling of satellite (OMI) measurements of HCHO columns. *Environmental Research Letters* **9(11)**, 114004 (2014).

REVIEWER COMMENTS

Reviewer #2 (Remarks to the Author):

The authors have made significant improvements in the revised manuscript. The evaluation with ground-based measurements made this work more robust. I also like the analysis of temporal variability.

I still have some minor comments.

1. It is not clear to me how the authors identified the hotspots. Do the authors apply certain thresholds? If so, please clarify the thresholds they use. The authors limit their analysis to 57 hotspots with "highest signal-to-noise ratios" in the second part. How do you define the signal to noise ratio here? Are these selected hotspots representative of the global hotspots? I'd suggest the authors show the distributions of these 57 selected hotspots in Figure 3.

2. Figure 3 shows a global distribution of C₂H₄ point sources, but is this a an exclusive list of all major C₂H₄ sources? I'd suggest the authors clarify in Figure 3 that these are just sources detectable from space.

Ethylene industrial emitters seen from space

Franco et al. - NCOMMS-22-11986

Response to the reviewers (R2)

Reviewer #2 (Remarks to the Author):

The authors have made significant improvements in the revised manuscript. The evaluation with ground-based measurements made this work more robust. I also like the analysis of temporal variability.

We thank the referee for the review of the paper and for the positive evaluation of the revised version of the manuscript. We have addressed (in blue) the new comments here below.

I still have some minor comments.

1. It is not clear to me how the authors identified the hotspots. Do the authors apply certain thresholds? If so, please clarify the thresholds they use. The authors limit their analysis to 57 hotspots with "highest signal-to-noise ratios" in the second part. How do you define the signal to noise ratio here? Are these selected hotspots representative of the global hotspots? I'd suggest the authors show the distributions of these 57 selected hotspots in Figure 3.

Regarding the identification of the hotspots: As the background levels of C₂H₄ vary greatly from one place to another, no automated way was found to identify the hotspots. In particular, the use of fixed thresholds was found to yield too many false detections. Therefore, we identified the hotspots based on a careful visual analysis of the hyperfine resolution map of C₂H₄ HRI together with satellite visible imagery and external sources (e.g., databases of chemical plants). We considered as hotspots those HRI enhancements that have a spatial extent of 20-50 km and in addition contrast well with the surrounding background HRI. Using the aforementioned external sources, we exclude certain hotspots related to e.g., fires or emissivity features. We now explain this better in the Methods.

"A thorough visual analysis of this distribution allowed the detection of 336 global hotspots (Fig. 3; Supplementary Tab. 1). These correspond typically to areas of 20-50 km spatial extent with HRI values significantly higher than the surrounding background. Additional satellite visible imagery and third-party sources were used to exclude false detections due to e.g., fires or emissivity features (see further)."

Regarding the 57 selected hotspots: Given the fact that the conversion from HRI to C₂H₄ columns introduces additional retrieval noise, we focused for the derivation of emission fluxes on those hotspots that stand out the most compared to the background. These are the largest hotspots but also some weaker hotspots observed in remote regions. Because the selection depends on the background levels, the list might not be fully representative of all the global hotspots (densely industrialized areas might be underrepresented because of elevated background concentrations making the detection of hotspots more challenging). However, in terms of magnitude of the emissions, as explained just above, the list contains both large and smaller sources (for instance, Asalouyeh, Iran, or Dahej, India, shown respectively in Figs 2 and 4). In addition, these 57 hotspots are distributed over the entire globe and cover all different types of identified sources (chemical, coal-related, metallurgy, etc.; see Fig. 3 here below). We have clarified this in the manuscript as follows:

"The retrieval of C₂H₄ total columns being challenging, we limit our estimates to the hotspots with the highest HRI values and to those presenting the largest contrasts relative to the surrounding background. In total, 57 global hotspots (53 related to industries and 4 to megacities) were quantified, representative of the different source categories and the global distribution of hotspots identified from space."

Following your suggestion, these 57 selected hotspots are now depicted as triangles in Fig. 3 (here below).

Figure 3 | Global distribution of C_2H_4 point-sources detected by IASI and their categorization. A total of 336 hotspots were identified. When a hotspot belongs to more than one category, only the two main categories are depicted on the map. The C_2H_4 emission fluxes are calculated for 57 hotspots represented as triangles.

2. Figure 3 shows a global distribution of C_2H_4 point sources, but is this an exclusive list of all major C_2H_4 sources? I'd suggest the authors clarify in Figure 3 that these are just sources detectable from space.

This is indeed a list of the C_2H_4 point-sources detected by IASI, and not an exhaustive list of all the C_2H_4 point-sources that exist worldwide. We have now made this clear in the caption of Fig. 3 (please, see the new version of Fig. 3 in our answer to the previous comment).

Ethylene industrial emitters seen from space

Franco et al. - NCOMMS-22-11986

Response to the reviewers (R3)

Reviewers' comments

No additional reviewers' comments.